# Widefield imaging of rapid pan-cortical voltage dynamics with an indicator evolved for one-photon microscopy

Xiaoyu Lu [1,7,10], Yunmiao Wang [2,8,10], Zhuohe Liu [3,9], Yueyang Gou [4], Dieter Jaeger [5,11] ✉ & François St-Pierre [1,3,4,6,11] ✉

Widefield imaging with genetically encoded voltage indicators (GEVIs) is a promising approach for understanding the role of large cortical networks in the neural coding of behavior. However, the limited performance of current GEVIs restricts their deployment for single-trial imaging of rapid neuronal voltage dynamics. Here, we developed a high-throughput platform to screen for GEVIs that combine fast kinetics with high brightness, sensitivity, and photostability under widefield one-photon illumination. Rounds of directed evolution produced JEDI-1P, a green-emitting fluorescent indicator with enhanced performance across all metrics. Next, we optimized a neonatal intracerebroventricular delivery method to achieve cost-effective and wide-spread JEDI-1P expression in mice. We also developed an approach to correct optical measurements from hemodynamic and motion artifacts effectively. Finally, we achieved stable brain-wide voltage imaging and successfully tracked gamma-frequency whisker and visual stimulations in awake mice in single trials, opening the door to investigating the role of high-frequency signals in brain computations.

Dissecting the neural underpinnings of behavior has been a long-pursued aim in neuroscience. Because many brain functions require coordinated computations between brain regions, they cannot be understood by studying brain areas in isolation. Therefore, capturing brain-wide activity and interactions is an increasingly common approach in systems neuroscience[1,2]. For example, monitoring large-scale neural activity can be used to determine the distributed representations of task features and decisions[3]. Several techniques have been applied for brain-wide recording of neural activity. NeuroPixel probes have been used to monitor activity from thousands of neurons across many brain regions[4,5]. However, these probes typically cannot distinguish between genetically defined cell populations, produce pan-cortical spatial activity maps, or report subthreshold membrane voltage fluctuations. Although spikes and local field potentials can be separated with silicon probe recordings, the latter arise from a complex mixture of sinks and sources from multiple cell types and layers[6].

Genetically encoded calcium indicators are alternative tools for monitoring neural activity with cell type specificity[7,8]. Calcium indicators have been deployed for brain-wide cortical imaging[9,10]. However, the slow indicator response kinetics and the intrinsically slow dynamics of calcium signals limit the ability of calcium indicators to follow rapid neural activity. Moreover, calcium indicators generally do

[1]Systems, Synthetic, and Physical Biology Program, Rice University, Houston, TX 77005, USA. [2]Neuroscience Graduate Program, Emory University, Atlanta, GA 30322, USA. [3]Department of Electrical and Computer Engineering, Rice University, Houston, TX 77005, USA. [4]Department of Neuroscience, Baylor College of Medicine, Houston, TX 77030, USA. [5]Biology Department, Emory University, Atlanta, GA 30322, USA. [6]Department of Biochemistry and Molecular Biology, Baylor College of Medicine, Houston, TX 77030, USA. [7]Present address: Department of Neuroscience, Baylor College of Medicine, Houston, TX 77030, USA. [8]Present address: Biology Department, Emory University, Atlanta, GA 30322, USA. [9]Present address: Department of Translational Molecular Pathology, The University of Texas MD Anderson Cancer Center, Houston, TX 77030, USA. [10]These authors contributed equally: Xiaoyu Lu, Yunmiao Wang. [11]These authors jointly supervised this work: Dieter Jaeger, François St-Pierre. ✉e-mail: djaeger@emory.edu; stpierre@bcm.edu

not report subthreshold depolarizations and hyperpolarizations[11,12]. Calcium indicators are thus poorly suited for reporting critical oscillatory properties of neural population activity, such as 15–30 Hz beta-band activity in the control of motor network function[13–15] and 40–70 Hz gamma oscillations that may dynamically link functional neural networks in perception and cognition[16–19].

Emerging tools for spiking and subthreshold electrical activity measurement are genetically encoded voltage indicators (GEVIs)— sensors that report voltage changes as variations in fluorescence[11]. GEVIs have been shown to report spikes, subthreshold depolarizations, and hyperpolarizations in genetically defined cell types and can feature faster temporal precision than calcium indicators[20–25]. Previous generations of GEVIs have been deployed for widefield cortical imaging[12,26–32]. However, while these GEVIs have been used for investigating slow cortical waves in the delta band (0.5–4 Hz)[29,32,33] they are suboptimal for following fast voltage changes such as gamma oscillations due to small or slow kinetics./or slow kinetics.

Here, we report on a GEVI optimized for one-photon (1P) imaging and improved methods for chronic brain-wide voltage imaging in behaving mice. We first discuss the development of an automated high-throughput platform for screening indicators across multiple performance characteristics. This system was utilized to evolve JEDI-1P, a GEVI with improved response amplitude, kinetics, brightness, and photostability. We optimized our genetic strategy to produce robust brain-wide expression of JEDI-1P and developed an optimized approach to correct hemodynamic and motion artifacts. The improvements in the indicator, expression strategy, and imaging methods enabled long-term brain-wide voltage imaging and successful tracking of high-frequency cortical voltage oscillations in the gamma band.

## Results

### Enabling multiparametric GEVI screening under widefield one-photon illumination

We first sought to develop a GEVI based on the ASAP family of indicators, where a circularly permuted GFP (cpGFP) is inserted in an extracellular loop of a voltage-sensing domain derived from a voltage-sensitive phosphatase[24]. Voltage-induced conformational changes in the voltage-sensing domain modulate the brightness of the coupled GFP (Fig. 1a). We chose to focus on this class of indicators because they have been shown to express efficiently and report rapid voltage dynamics in a variety of neuronal types and experimental preparations[24,25,34–37].

Indicator properties—response amplitude, kinetics, brightness, and photostability—can depend on the illumination modality[38–40]. We sought to develop a screening platform to optimize GEVIs for widefield (one-photon) imaging, in contrast to our parallel efforts to improve GEVIs for laser-scanning two-photon microscopy[41]. Prior screens show that while mutations can enhance one performance metric, they may degrade others that weren't measured[21]. To achieve multiparametric or 'holistic' consideration of a GEVI's performance, we designed a platform to screen for all fundamental indicator properties in the same assay—response amplitude, kinetics, brightness, and photostability.

The first performance metric we considered was the ability of GEVIs to report rapid voltage transients such as action potentials (APs) and post-synaptic potentials. We fitted a widefield inverted fluorescence microscope with a camera capable of capturing fluorescence with a temporal resolution of 1 ms per image in a field-of-view of 2048 × 200 pixels (Fig. 1b). We screened indicators using a HEK293 cell line that has a resting membrane potential around −77 mV, similar to that of adult mammalian cortical neurons[34,42,43]. 100-ms electric field stimulation (EFS) pulses were previously reported to depolarize these cells[21,44] (Supplementary Fig. 1a). To benchmark GEVIs' ability to report faster voltage signals, we shortened the pulse duration to 1 ms (Fig. 1c). The GEVIs responded to the pulses, exhibiting transients with

full-width at half-maximum of 11-22 ms, depending on indicator kinetics (Supplementary Fig. 1b). Response amplitudes to AP waveforms and 1-ms EFS pulses were strongly correlated (Fig. 1d), suggesting that our platform could screen for larger responses to short voltage transients.

We also aimed to identify slower indicators with enhanced responses to longer voltage signals and faster variants with diminished responses, aiming to combine their mutations. We observed that the peak responses to 100-Hz trains of ten 2.5-ms EFS pulses were highly correlated with the peak responses to 1-s 100-mV step depolarizations (Fig. 1e). The width of AP waveforms and EFS-induced spikes were highly correlated (Fig. 1f). These results demonstrate we can rank indicators based on their response amplitudes to longer voltage signals and their kinetics.

We also evaluated indicators for photostability (Fig. 1g) and brightness (Fig. 1h). To normalize brightness measurements for variations in expression level, we co-expressed the red fluorescent protein mCherry using a self-skipping 2A sequence[45] (Fig. 1h). mCherry was anchored to the plasma membrane via a CAAX membrane targeting motif[46]. As expected, the GEVI/mCherry ratio showed less variation between fields-of-view compared with GEVI (green) intensity values alone (Supplementary Fig. 1c). Finally, to maximize screening throughput, we excluded wells that did not meet pre-defined quality control standards from further data collection (Supplementary Fig. 1d).

### Multiparametric screening identified JEDI-1P, an indicator with faster kinetics, higher response amplitudes, and a larger photon budget under one-photon illumination

We applied our platform to screen single-site saturation mutagenesis libraries in 96-well plates. ASAP1[24] and ASAP2s[34] were selected as starting templates, since ASAP3[25] had not been reported when this project started. We targeted residues around the chromophore of cpGFP, in conserved positions of the voltage-sensing domain, and near the junctions between cpGFP and the voltage-sensing domain. We screened over 1,200 mutants from 21 single-site libraries (Fig. 1i), and promising mutations were combined (Fig. 1j–l and Supplementary Fig. 1e–g). Our best variant includes 5 mutations from ASAP2s. We named this sensor Jellyfish-derived Electricity-reporting Designer Indicator for 1-Photon, or JEDI-1P.

We evaluated the responses of JEDI-1P to voltage changes in single human cells (HEK293A) using one-photon widefield imaging and whole-cell voltage clamp. JEDI-1P produced steep responses between −80 and 0 mV, demonstrating it is well adapted to reporting voltage signals within the physiological range of cortical neurons (Fig. 2a, b). JEDI-1P produced larger responses to 100-mV (−70 to +30 mV) step depolarizations compared with ASAP2s and ASAP3 (Fig. 2c). JEDI-1P responded maximally to voltage changes at physiological pH (Supplementary Fig. 2). At -33 °C, JEDI-1P demonstrated on- and off- kinetics of $0.54 \pm 0.07$ and $1.20 \pm 0.12$ ms (mean ± 95% confidence interval here and henceforth), respectively, faster than ASAP2s and ASAP3 (Table 1).

With its larger response amplitudes and faster kinetics, JEDI-1P responded to spike waveforms with change of $−31.2 \pm 2.0\%$, corresponding to a 174% increase over the ASAP2s' response of $−11.4 \pm 0.8\%$ and a 71% increase over ASAP3's response of $−18.3 \pm 2.0\%$ (Fig. 2d, e). JEDI-1P consistently produced larger responses to spike waveforms with widths ranging from 0.25 to 4 ms (Supplementary Fig. 3a, b). Due to its faster kinetics, JEDI-1P fluorescence returned closer to the baseline between individual spikes of a train of AP waveforms (Fig. 2d) and produced narrower responses to spike waveforms (Fig. 2f). JEDI-1P also responded to high-frequency spiking of Purkinje-like AP waveforms with greater precision than ASAP2s and ASAP3 (Supplementary Fig. 3c–e).

Our next objective was to compare the brightness and photostability of JEDI-1P, ASAP2s, and ASAP3. When exposed to illumination

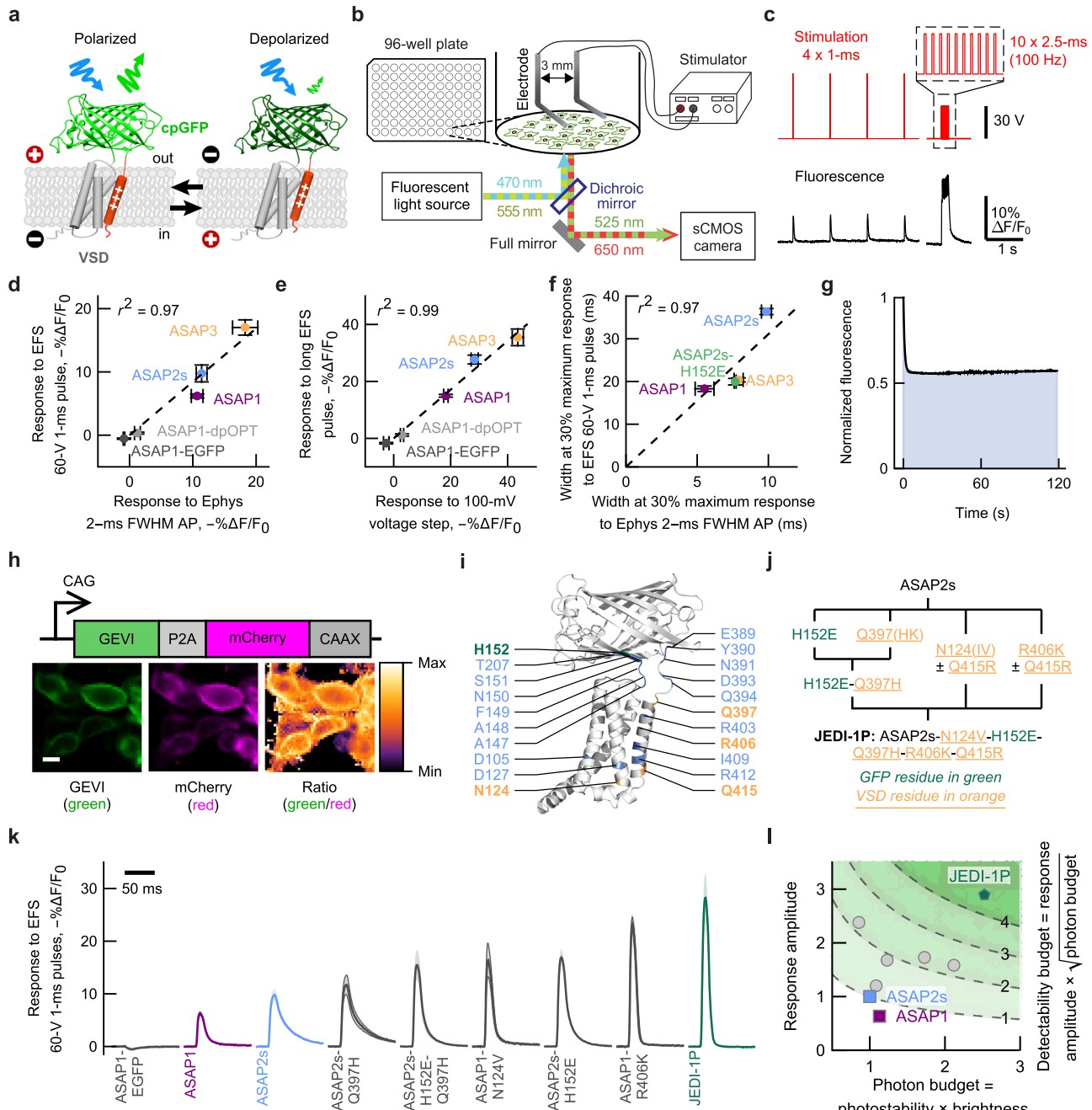

**Fig. 1 | Multiparametric one-photon screening identified JEDI-1P, a voltage indicator with improved performance. a** We engineered GEVIs composed of a circularly permuted green fluorescent protein (cpGFP, green) whose brightness is modulated by voltage-induced conformational changes in the voltage-sensitive domain (VSD, gray and orange). **b** Schematic of the platform for GEVI screening under widefield one-photon illumination. **c** *Top*, Electric field stimulation (EFS) protocol. Inset: zoom-in of the 100-Hz pulse train. *Bottom*, representative fluorescence responses of a parental indicator, ASAP1. **d** Fluorescence responses to 1-ms EFS pulses and AP-like waveforms (2-ms width at half-maximum, +30 mV peak, −70 mV baseline, whole-cell voltage clamp) are highly correlated. For (**d**–**f**), $n = 4$ independent transfections per GEVI (EFS) or 3 (ASAP1-dpOPT), 4 (ASAP1-EGFP), 4 (ASAP1), 10 (ASAP2s), 4 (ASAP2s-H152E), and 8 (ASAP3) HEK293A cells (voltage clamp). **e** Fluorescence responses to EFS pulse trains and 1-s 100-mV step depolarizations (−70 mV to + 30 mV, whole-cell voltage clamp) are highly correlated. **f** Optical response widths to 1-ms EFS pulses and action potential (AP)-like

waveforms are highly correlated. Widths were measured at 30% of the response peak. **g** Photostability was quantified as the area under the curve of fluorescence vs. time graphs (shaded area in this example photobleaching curve). Fluorescence was normalized to its value at $t = 0$. The irradiance was ~33 mW/mm². **h** Brightness was evaluated as the GEVI/mCherry (green/red) fluorescence ratio. Scale bar, 10 μm. **i** The 21 locations targeted by mutagenesis are indicated on a predicted 3D structure of ASAP2s. Green and orange residues were mutated in JEDI-1P. **j** Key steps of the evolutionary path leading to JEDI-1P. VSD mutations were screened under both 415R (ASAP1) and 415Q (ASAP2s) contexts. **k** JEDI-1P shows faster kinetics and a higher response amplitude to 1-ms EFS pulses than parental indicators (ASAP1 and ASAP2s) and screening intermediates. **l** GEVI multiparametric directed evolution. Values are relative to those of ASAP2s. Gray circles are screening intermediates. All panels: Error bars or shading denote the 95% confidence interval. Pearson's $r^2$ was calculated from intercept-free unweighted linear regression of the mean values.

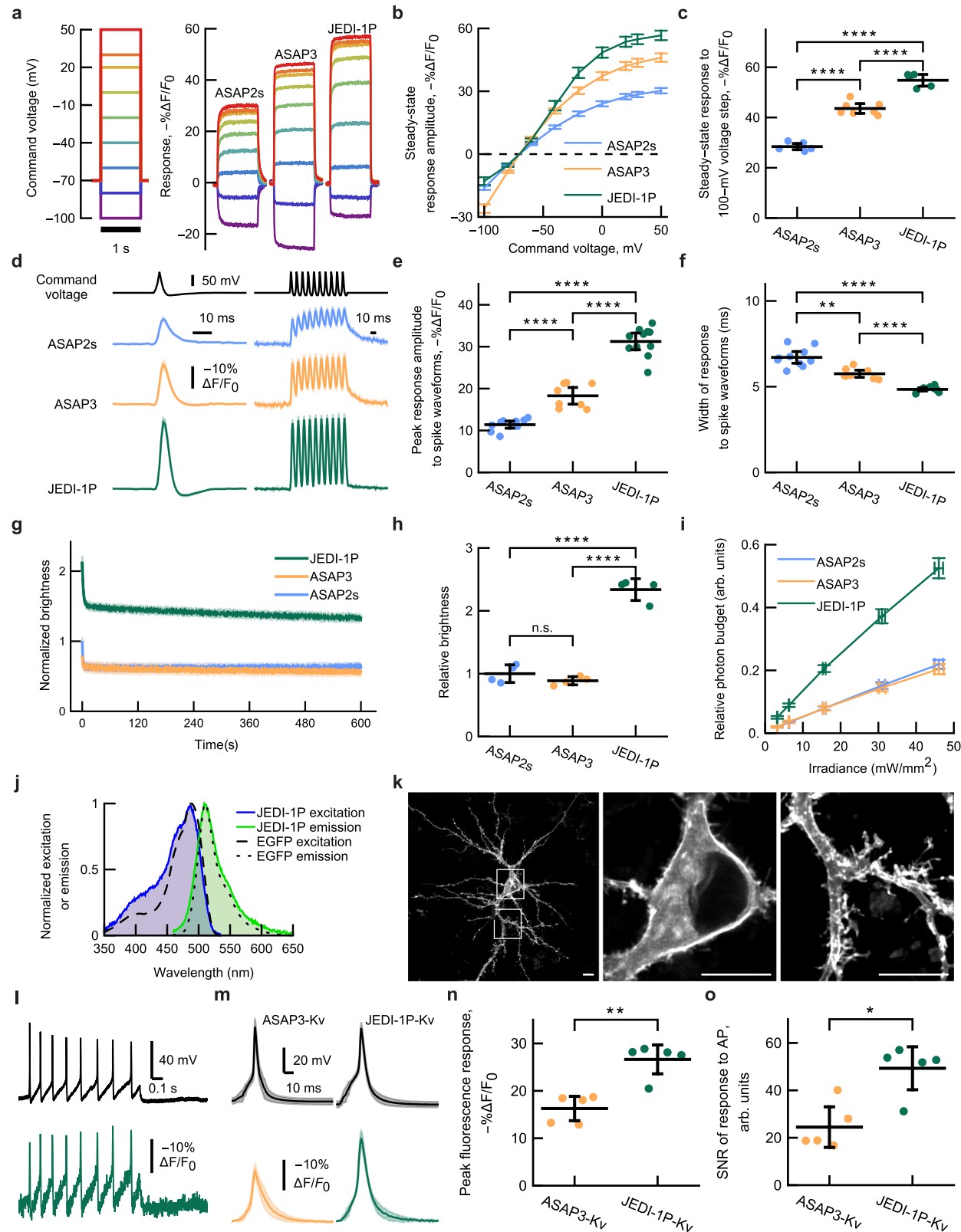

at ~3 mW/mm², the three indicators exhibited high photostability after an initial ~5-s period of rapid photobleaching (Fig. 2g). This result suggested that these GEVIs would exhibit high photostability during in vivo widefield imaging, where the illumination irradiance is often over 60 times weaker. Compared with ASAP3 and ASAP2s, JEDI-1P had higher brightness at the beginning and throughout 2-5 min trials (Fig. 2g, h, and Supplementary Fig. 4). As a result of its

high photostability and greater brightness, JEDI-1P exhibited a larger photon budget (area under the curve of the relative brightness trace) at all irradiance levels tested (Fig. 2i). To help determine optimal voltage imaging parameters, we acquired JEDI-1P's excitation and emission spectra. JEDI-1P was maximally excited at 487 nm and produced an emission spectrum with a peak at 509 nm, similar to EGFP (Fig. 2j). JEDI-1P can thus be imaged under similar optical

**Fig. 2 | JEDI-1P has improved sensitivity, kinetics, brightness, and photo-stability under one-photon illumination in vitro. a–c** JEDI-1P has greater steady-state response amplitude to 1-s step depolarizations than ASAP2s and ASAP3. Fluorescence was acquired at ~1 kHz. $n = 6$ (ASAP2s), $n = 7$ (ASAP3) and $n = 5$ (JEDI-1P) HEK293A cells. **a** Mean response to voltage steps. **b** Quantification of (**a**). **c** Responses to 100-mV step voltages (−70 mV to +30 mV). $p = 3.6 \times 10^{-11}$ (One-way ANOVA). **d–f** JEDI-1P shows larger and faster responses to short voltage transients compared with ASAP2s and ASAP3. Fluorescence was acquired at 1 kHz. $n = 10$ (ASAP2s), $n = 8$ (ASAP3), and $n = 11$ (JEDI-1P) HEK293A cells. **d** *Left*: mean response to spike waveforms simulating APs (2-ms full-width at half-maximum, +30 mV peak, baseline at −70 mV). *Right*: mean response to a 100-Hz spike train waveform. **e** JEDI-1P has a larger peak response amplitude to single AP waveforms than ASAP2s and ASAP3. $p = 7.0 \times 10^{-15}$ (One-way ANOVA). **f** JEDI-1P responds more quickly to single AP waveforms than ASAP2s and ASAP3, as shown by narrower optical responses. The width corresponds to the full-width at half-maximum. $p = 8.4 \times 10^{-8}$ (Welch's One-way ANOVA). Pairwise tests with Dunnett's T3 multiple comparisons test (two-tailed). **g** 10-min fluorescence trace of JEDI-1P, ASAP2s, and ASAP3 illuminated with 470/24-nm light at an irradiance of 3.2 mW/mm² at the sample plane, normalized to the brightness of ASAP2s in the first frame. $n = 8$ independent transfections per GEVI. **h** JEDI-1P is brighter than ASAP2s and ASAP3. Brightness was quantified as described in Fig. 1h. $n = 4$ independent transfections per GEVI. $p = 1.7 \times 10^{-7}$ (One-way ANOVA). **i** JEDI-1P displays larger relative photon budget (area under the curve

of the normalized brightness, timecourses of 2 min) than ASAP2s and ASAP3 over a > 10-fold range of irradiance levels (3.2–46 mW/mm²). $n = 12$ (ASAP2s), $n = 11$ (ASAP3) and $n = 12$ (JEDI-1P) independent transfections per GEVI. **j** JEDI-1P has an excitation peak at 487 nm and an emission peak at 509 nm. $n = 6$ (EGFP) and 5 (JEDI-1P) independent transfections in HEK293-Kir2.1 cells. **k** JEDI-1P efficiently traffics to the plasma membrane in the soma and dendrites. Representative confocal image acquired from a DIV14 rat cortical neuron. Scale bars, 10 μm. **l** Single-trial recording of a representative dissociated cortical neuron (DIV12) expressing JEDI-1P-Kv. *Upper*: membrane voltage recording from whole-cell current clamp in response to a 100-pA current injection. *Lower*: voltage imaging of JEDI-1P. **m–o** JEDI-1P-Kv detects APs with larger response amplitudes and greater signal-to-noise ratio (SNR) than ASAP3-Kv in dissociated hippocampal neurons (DIV14-16). $n = 5$ neurons per variant. Experiments were done at 32–35 °C. **m** Mean fluorescence response (lower panel) to current-triggered APs (upper panel). Triggering was performed using 50–400 pA injected over 20 ms. **n** JEDI-1P-Kv produced larger fluorescence responses to APs than ASAP3-Kv. $p = 0.0079$ (Mann–Whitney $U$ test, two-tailed). JEDI-1P-Kv's responses exhibited a higher SNR for AP detection than ASAP3-Kv. $p = 0.0159$ (Mann–Whitney $U$ test, two-tailed). All panels: ****$p < 0.0001$; ***$p < 0.001$; **$p < 0.01$; *$p < 0.05$; n.s. $p > 0.05$. Tukey's HSD multiple comparison test (two-tailed) was used unless otherwise noted. Error bars or shading denote the 95% confidence interval (CI) of the mean. Experiments were done at room temperature unless otherwise noted.

configurations as popular GFP-based sensors such as calcium indicators of the GCaMP family[7].

The development of JEDI-1P for one-photon microscopy was concurrent with the engineering of JEDI-2P, a GEVI optimized for two-photon (2P) illumination that we recently reported[41]. We sought to compare these two indicators under 1P and 2P. Using widefield 1P illumination, JEDI-2P—but not JEDI-1P—displayed a decrease in response amplitude over time (Supplementary Fig. 5). This observation suggests that our 1P illumination conditions convert JEDI-2P molecules to a fluorescent but voltage-unresponsive state. We speculate that JEDI-2P's T207H mutation, also present in photoactivatable GFP[47], may be responsible for this behavior. Under laser-scanning 2P, JEDI-1P displayed similar properties to JEDI-2P except for a lower photostability (Supplementary Fig. 6).

Finally, before conducting in vivo experiments, we characterized JEDI-1P in dissociated neurons. We first confirmed that JEDI-1P could be expressed and efficiently trafficked to the plasma membrane in the soma and dendrites of cortical neurons in vitro (Fig. 2k), similar to its parental sensor ASAP2s[34]. JEDI-1P detected AP trains and subthreshold depolarizations in single trials (Fig. 2l, Supplementary Fig. 7a). JEDI-1P reported APs with higher sensitivity than ASAP3, as indicated by larger response amplitudes and signal-to-noise ratios (SNRs) (Fig. 2m–o, Supplementary Fig. 7b).

## Robust pan-cortical expression of soma-targeted JEDI-1P

Because of its improved ability to report voltage signals, we selected JEDI-1P to conduct pan-cortical imaging experiments. To preferentially capture somatic voltage signals, JEDI-1P was appended with a motif from the potassium channel Kv2.1, which restricts membrane protein expression to the soma and proximal dendrites[48]. The resulting variant, JEDI-1P-Kv, was cloned in a Cre-dependent expression vector. JEDI-1P-Kv adeno-associated viruses (AAVs) were packaged using PHP.eB, a capsid that produces efficient gene transfer across the central nervous system[49,50]. AAVs were injected in *EMX1-cre* mice to obtain preferential JEDI-1P-Kv expression in cortical excitatory neurons[51,52].

Total hemoglobin fluctuations and differential light absorption of oxygenated and deoxygenated hemoglobin produce artifactual signals during widefield imaging of neural activity indicators[53–59]. Imaging a reference fluorescence signal at a different emission wavelength is a common and effective approach to isolate neural signals from hemodynamic activity[56,58–60]. Therefore, to correct JEDI-1P-Kv measurements from hemodynamic and motion artifacts, we co-injected AAVs expressing the red fluorescent protein (RFP) mCherry from the pan-

neuronal promoter *hSyn* (or tdTomato from the non-cell-type-specific promoter *CAG*).

We chose to administer neonatal intracerebroventricular (ICV)[61] over retro-orbital[49] injections, given the 40-fold higher cost of the latter method. To facilitate imaging, we employed a clear-skull preparation that preserves skull integrity[62,63]. Four weeks after neonatal ICV injections, we observed strong JEDI-1P-Kv expression throughout the cortex and in *EMX1*-expressing subcortical structures such as the hippocampus (Fig. 3a). As expected, JEDI-1P-Kv expression was limited to somatic and proximal dendritic membranes. Cytosolic mCherry or td-Tomato were also expressed cortex-wide (mCherry: Fig. 3a, right; tdTomato, Supplementary Fig. 8g). We imaged the intact cortex of head-fixed mice at 200 Hz (Fig. 3b). While there were differences in fluorescence across the cortex and between individuals, JEDI-1P-Kv could be reliably detected above background fluorescence (Fig. 3c-d). Reliable mCherry signals were obtained with the same 466/40-nm-filtered blue excitation light source we used to image JEDI-1P-Kv (Fig. 3e). Using a single light source eliminated the need for temporal switching between channels, which allowed for rapid voltage imaging. Additionally, illumination noise could be more easily corrected as it simultaneously affected JEDI-1P-Kv and reference channels.

## JEDI-1P-Kv is photostable during widefield longitudinal imaging experiments

GEVI photobleaching can severely limit the duration of voltage imaging sessions with some indicators and experimental preparations[11,21]. However, we observed minimal photobleaching with JEDI-1P-Kv during widefield imaging (Fig. 3f–i). Although the raw fluorescence intensity decayed to $94.1 \pm 0.3\%$ (represented as mean ± 95% CI here and henceforth) by the end of a 20-s trial (Fig. 3f), the fluorescence level recovered almost entirely during the 20-s dark intervals between trials, leading to $98.4 \pm 1.3\%$ intensity remaining at the end of an imaging session (Fig. 3g). After more than 5 h of imaging over 19 days—10 imaging sessions of one hundred 20-s trials—JEDI-1P-Kv conserved $77.4 \pm 1.3\%$ of its fluorescence (Fig. 3h). Under continuous illumination, JEDI-1P-Kv's intensity decreased to $95.8 \pm 4.2\%$ after 1 h (Fig. 3i). Consistent with our 466/40-nm light source exciting mCherry far from its absorption peak of 587 nm[64], mCherry displayed even higher photostability than JEDI-1P-Kv. It exhibited $92.8 \pm 3.3\%$ of its initial fluorescence after the last imaging session and $100.2\% \pm 2.5\%$ after 1 h of continuous illumination. Overall, the excellent photostability of JEDI-1P-Kv, as observed with our widefield system, enabled repeated and longitudinal in vivo optical recordings with a high SNR.

**Table 1 | JEDI-1P is faster than ASAP2s and ASAP3, related to Fig. 2**

| | JEDI-1P | ASAP3[a] | ASAP2s[a] |
|---|---|---|---|
| *Depolarization time constants at 32–35 °C* | | | |
| $\tau_{fast}$ (ms) | 0.54 ± 0.07 | 0.98 ± 0.15 | 3.78 ± 0.22 |
| $\tau_{slow}$ (ms) | 10.3 ± 1.8 | 20.9 ± 16.2 | 132 ± 63 |
| Proportion fast (%) | 89.3 ± 2.3 | 82.4 ± 5.9 | 93.6 ± 1.2 |
| *Repolarization time constants at 32–35 °C* | | | |
| $\tau_{fast}$ (ms) | 1.20 ± 0.12 | 4.29 ± 0.68 | 7.59 ± 0.60 |
| $\tau_{slow}$ (ms) | n/a[b] | 136 ± 47 | 170 ± 25 |
| Proportion fast (%) | n/a[b] | 86.9 ± 11.2 | 91.7 ± 0.6 |
| *Depolarization time constants at 21–23 °C* | | | |
| $\tau_{fast}$ (ms) | 1.74 ± 0.12 | 2.26 ± 0.55 | 6.39 ± 0.34 |
| $\tau_{slow}$ (ms) | 27.8 ± 3.0 | 32.2 ± 18.8 | 125 ± 43 |
| Proportion fast (%) | 85.9 ± 1.4 | 78.4 ± 11.0 | 91.3 ± 1.3 |
| *Repolarization time constants at 21–23 °C* | | | |
| $\tau_{fast}$ (ms) | 2.91 ± 0.17 | 11.0 ± 1.2 | 12.7 ± 0.9 |
| $\tau_{slow}$ (ms) | n/a[b] | 112 ± 44 | 143 ± 22 |
| Proportion fast (%) | n/a[b] | 85.3 ± 10.9 | 87.1 ± 2.2 |

Kinetics were measured in response to 1-s voltage steps in HEK293A cells under whole-cell voltage clamp. Cells were clamped at −70 mV at rest. Depolarization was performed from −70 mV to 30 mV while repolarization was performed from 30 mV to −70 mV. Values of JEDI-1P are means ± 95% CI from *n* = 9 cells at 21–23 °C and *n* = 10 cells at 32–35 °C.
[a]Values from ref. 41. Experiments were performed under the same condition as JEDI-1P. Values are means ± 95% CI from *n* = 9 cells (ASAP2s and ASAP3) at 21–23 °C and *n* = 10 cells (ASAP2s), *n* = 16 cells (ASAP3) at 32–35 °C.
[b]These kinetics were best fit by a monoexponential function.

## Data pre-processing pipeline isolates JEDI-1P-Kv voltage signals from hemodynamic artifacts

To enhance the accuracy of voltage signals recorded with JEDI-1P-Kv, we devised a pre-processing pipeline that corrects autofluorescence and photobleaching and removes hemodynamic and motion artifacts. Autofluorescence was subtracted using values obtained from uninjected control mice. Single-trial time courses were corrected for photobleaching using a bi-exponential fit. The remaining signal had an apparent heartbeat artifact (Fig. 4a), which was also present in the RFP channel. The heartbeat artifact was successfully removed from the voltage signal by regressing the RFP channel filtered around the heartbeat frequency (Fig. 4a, b). We corrected signal artifacts most effectively by regressing three frequency bands since contaminating signal amplitudevaried with frequency (Fig. 4c, d, Supplementary Fig. 8a–f and Fig. 9).

The number of regression steps and their respective bandwidth can be customized based on the motion and hemodynamic characteristics of the desired preparation, such as anesthesia or wakefulness. The pipeline produced similar results using data acquired with tdTomato rather than mCherry (Supplementary Fig. 8g–i). Overall, pixel-wise regression can be an effective method to remove hemodynamics[59], and our pipeline using sequential regression at different frequency ranges provides a flexible strategy to regress artifacts.

## JEDI-1P-Kv signals correlate with local field potentials at frequencies up to 60 Hz

To investigate the relation between JEDI-1P-Kv fluorescence and local field potentials (LFP), we conducted concurrent widefield imaging and acute recordings of barrel cortex LFPs in mice lightly anesthetized using isoflurane (Fig. 5a). Single-trial optical recordings closely tracked LFPs recorded near the imaging site (Fig. 5b). We quantified the correlation between barrel cortex LFPs and JEDI-1P-Kv signals across the entire cortical area. We found that, unsurprisingly, LFP signals were most correlated with optical responses near the LFP recording site

(Fig. 5c, d). We observed a network spanning bilateral barrel and motor cortices with voltage dynamics correlated with barrel cortex LFPs. This finding is consistent with previous studies that demonstrated shared LFP components across functional networks using fMRI and widefield Ca2+ imaging[56,65,66]. We did not expect correlation coefficients to approach the maximum value of 1.0 since LFP recordings were not limited to excitatory neurons, corresponded to neurons located in different locations and depths, and represented a complex mixture of local potentials and volume conductance from distant sources[67].

We used coherence analysis to investigate the relationship between LFPs and JEDI-1P-Kv signals at different frequencies. We found that anterior ROIs showed a clear coherence peak around 8 Hz, which was stronger on the ipsilateral side and absent in posterior ROIs (Fig. 5e). The analyzed site closest to the LFP recording showed coherence up to ~50 Hz, while coherence with the ipsilateral motor cortex extended to 40 Hz. Coherence with contralateral structures was most dominant at frequencies below 8 Hz. When we recorded optical and LFP signals simultaneously in awake mice exposed to sensory stimulation, coherence was observed up to ~60 Hz (Supplementary Fig. 10). These findings demonstrate the ability of JEDI-1P-Kv widefield imaging to reveal functional networks across different frequency bands up to 60 Hz, a valuable feature that has not been demonstrated with other voltage indicators.

## JEDI-1P-Kv signals track cortical voltage responses to whisker and visual stimulation trains up to 60 Hz

We next investigated whether JEDI-1P-Kv's rapid kinetics would enable the monitoring of fast neuronal responses to sensory stimulation under widefield imaging in vivo. We unilaterally applied single whisker deflection on the C2 whisker of awake mice while imaging cortical activity (Fig. 6a). JEDI-1P-Kv rapidly responded to single whisker deflections in the motor and contralateral primary sensory cortices (Fig. 6b, c and Supplementary Fig. 11), consistent with prior studies using voltage-sensitive dyes[68]. We also applied 20-60 Hz visual flickers to awake mice during imaging (Fig. 6d), and determined which cortical areas showed voltage fluctuations following the stimulation frequency. The bilateral visual cortex produced the strongest response at the stimulation frequencies (Fig. 6e).

To assess our ability to monitor the response of small cortical areas to different stimuli, we quantified JEDI-1P-Kv signals from a 2 × 2-pixel area (200 × 200 μm) in the visual cortex. Single-trial spectrograms revealed clear signals at the frequencies matching the ongoing visual flicker (Fig. 6f and Supplementary Fig. 12a). Correspondingly, voltage traces acquired during visual stimuli presentation showed oscillations, indicating that excitatory visual cortex neurons could follow high-frequency visual inputs (Fig. 6g).

To determine whether fast voltage responses to somatosensory stimuli could be detected, we conducted widefield voltage imaging in response to 25–60 Hz air-puff trains (Fig. 6h–k and Supplementary Fig. 12b). We filtered the responses to monitor those matching the stimulus frequency and found strong responses in the contralateral barrel cortex. Interestingly, we also observed robust responses in the contralateral retrosplenial cortex (RSPd) (Fig. 6i), an area commonly associated with spatial navigation[69] and receiving input from the barrel cortex[70]. Fluorescence responses remained easily detectable over many recording sessions (Fig. 6l, m), highlightingJEDI-1P-Kv's potential for reporting fast network dynamics inlongitudinal studies.

## Discussion

Behavioral outcomes are rooted in neural computations that activate specific brain regions, necessitating large-scale and precise recording technologies. Here, we addressed this critical need by engineering an enhanced GEVI and devising optimized widefield imaging methods for pan-cortical voltage recordings with high spatiotemporal resolution.

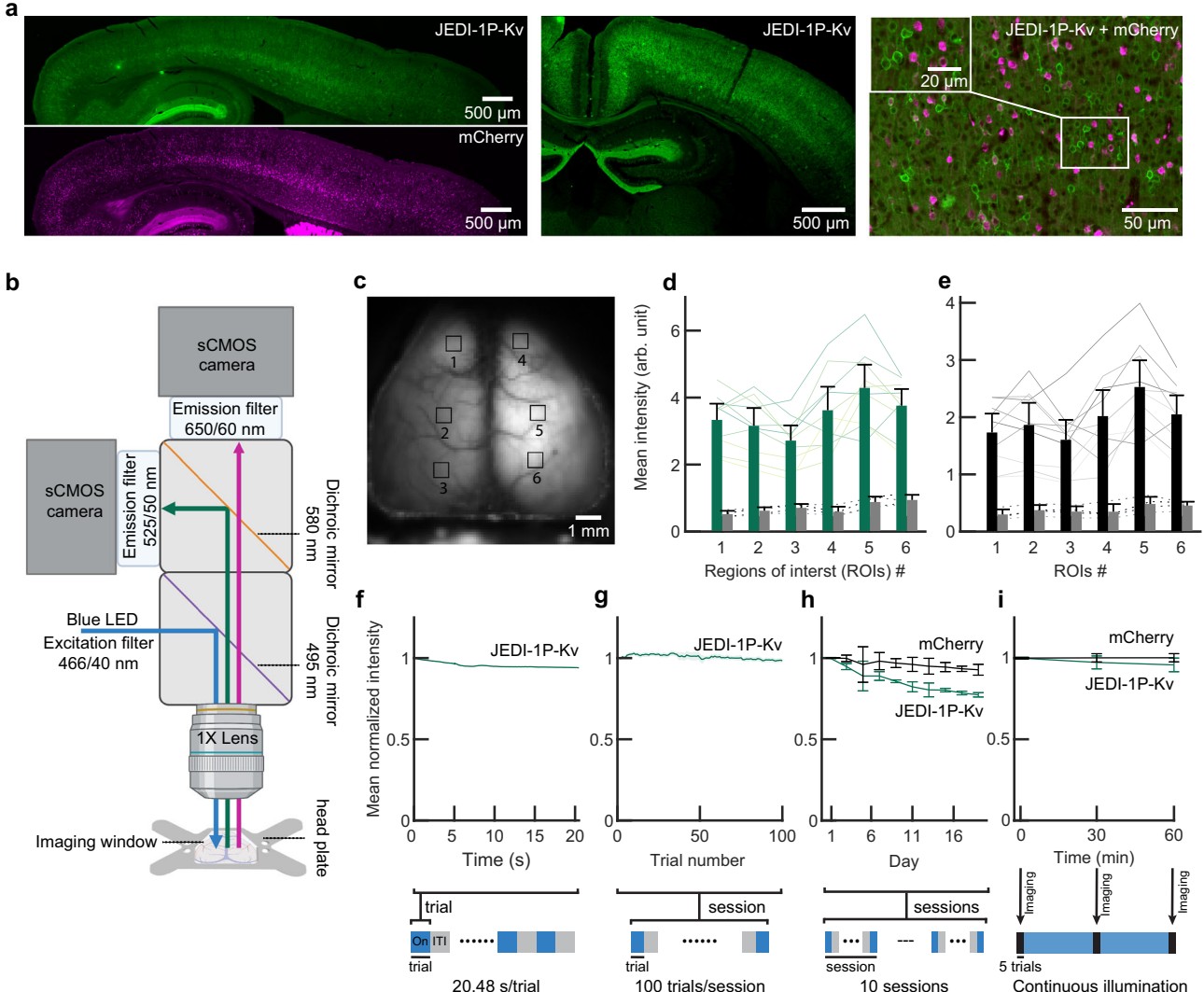

**Fig. 3 | Reliable pan-cortical expression of JEDI-1P-Kv. a** Sagittal slide illustrating that JEDI-1P-Kv (*top left*) and mCherry (*bottom left*) are expressed pan-cortically. *Middle*, coronal section. *Right*, zoomed-in and overlaid image of JEDI-1P-Kv and mCherry. Adeno-associated viruses encoding soma-localized JEDI-1P (JEDI-1P-Kv) and a cytosolic red-emitting reference fluorescent protein (mCherry) were introduced bilaterally by neonatal intracerebral ventricular (ICV) injection. **b** Imaging setup schematic. All experiments were conducted at 200 Hz. **c** Representative image of pan-cortical JEDI-1P-Kv fluorescence acquired with the system depicted in (**b**). The numbered squares show the 6 × 6-pixel (600 × 600 μm) regions of interest (ROIs) analyzed in (**d**) and (**e**). **d, e** JEDI-1P-Kv (**d**) and mCherry (**e**) can be robustly detected over the background. Bars represent the mean fluorescence intensity at different ROIs from mice expressing JEDI-1P-Kv and mCherry (green and black bars, respectively; *n* = 10) or control mice expressing neither (gray bars, *n* = 4). Lines show data from individual mice expressing JEDI-1P-Kv or mCherry (solid) or control mice (dashed). Mean intensities of all ROIs from mice expressing JEDI-1P-Kv and

mCherry are significantly higher than those from control mice. *p* = 0.002 for each ROI, two-sided Mann–Whitney *U* test. **f–i** JEDI-1P-Kv is photostable under wide-field illumination in vivo. Excitation power was 0.05 mW/mm², lower than used in vitro (Fig. 2g, h). Data shows mean JEDI-1P-Kv fluorescence intensity over a single trial (**f**), a daily session (**g**), all 10 imaging sessions (**h**), and 1-h continuous illumination (**i**). Bottom schematics illustrate the illumination programs. Inter-trial intervals (ITI) were ~20 s. Total imaging time per session was 34 min. During continuous illumination, images were acquired at 0, 30, and 60 min; 5 trials/animal at each timepoint. Data were normalized to the beginning of the trial (**f**), the mean intensity of the first trial (**g**), the mean intensity of the first session of the first day (**h**), and the mean intensity at 0 min (**i**). *n* = 300 trials from 3 mice (100 trials/animal) (**f**), 27 sessions from 3 mice (**g**), 3 mice (**h**), and 4 mice (**i**). For (**d–i**), mean intensities are shown. Error bands (**f**, **g**) and bars (other panels) denote the 95% CI. Panel (**b**) was created using Biorender.com.

To achieve robust voltage imaging, we developed an automated multi-parametric screening platform to evolve GEVI variants under widefield one-photon illumination. Our system can evaluate a variant's brightness, photostability, kinetics, and voltage sensitivity (Fig. 1). Our platform accelerated the screening process while avoiding outcomes where improvements in certain traits were accompanied by decreases in others[34]. The overall best indicator, JEDI-1P, outperforms previous sensors: it is brighter, more photostable, and responds more quickly and robustly to voltage transients (Fig. 2). JEDI-1P is preferred over JEDI-2P, a 2P-optimized indicator[41], for 1P widefield experiments. (Supplementary Fig. 5). However, JEDI-1P was more photolabile than JEDI-2P

under laser-scanning 2P microscopy (Supplementary Fig. 6). These findings underscore the need for customized screening pipelines to develop indicators optimized for specific illumination modalities.

JEDI-1P's upgraded features enabled 200-Hz voltage imaging through a cleared skull with high SNR (Figs. 3d, 4a, 5a, b, 6) and minimal photobleaching (Fig. 3f–h). At ~34 °C, JEDI-1P exhibits ~0.5 and ~1.2-ms on- and off-kinetics, respectively (Table 1), enabling single-trial reporting of voltage transients with comparable timescales as LFPs (Fig. 5b, e) and neural responses to stimuli up to 60 Hz (Fig. 6).

We also reported approaches to promote the rapid adoption of widefield voltage imaging by the neuroscience community. First, we

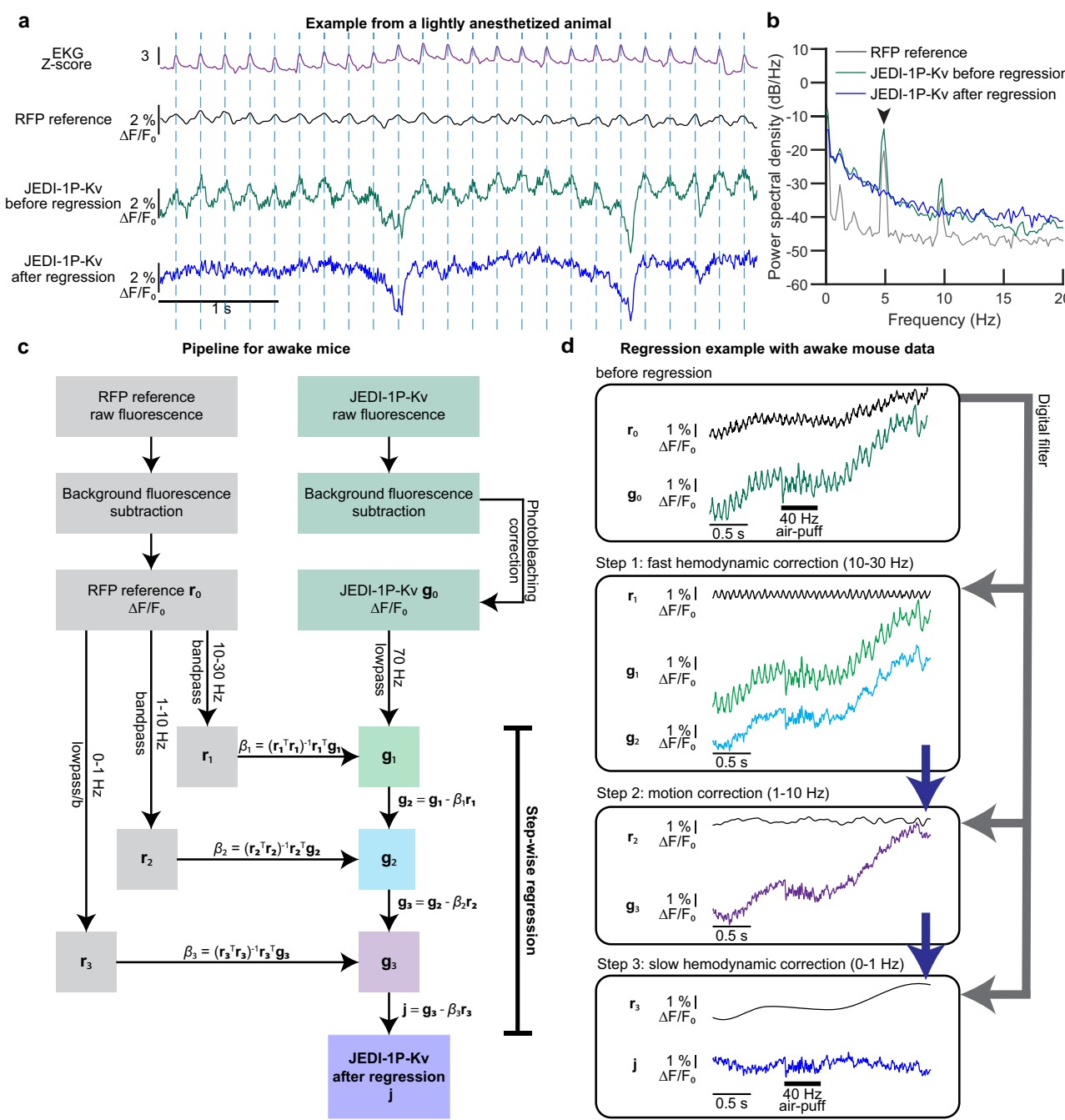

**Fig. 4 | Hemodynamic and motion correction can be achieved with a red fluorescent protein and a single excitation light source. a** 5-Hz oscillations observed in JEDI-1P-Kv and RFP reference channels in lightly anesthetized mice correspond to the heart rate (shown in EKG) and reflect fast blood volume changes (fast hemodynamics). JEDI-1P-Kv and RFP optical traces are from the same 2 × 2-pixel ROI. The regression pipeline for anesthetized mice is as in panel (**c**) but used 4–20, 1–4, 0–1 Hz as the three bandpass filters. **b** Successful removal of fast hemodynamic signals in a lightly anesthetized animal. *n* = 25 trials from a representative mouse. The heartbeat peak at ~5 Hz (arrow) was removed in the regressed trace (blue). **c** Signal pre-processing workflow when recording from awake animals. The reference channel was filtered to three different frequency ranges for regression: 10–30 Hz—which includes the heartbeat frequency in the awake condition, 1–10 Hz, and 0–1 Hz. These ranges produced efficient regression in our awake mice, but different ranges might be optimal in other studies. **d** Step-wise regression on representative traces removed fluorescence signals unrelated to voltage. An example of pixel-wise regression is shown in Supplementary Fig. 9.

demonstrated that soma-restricted JEDI-1P can be broadly and robustly expressed in vivo following ICV viral injections (Fig. 3), reducing costs by over an order of magnitude over retro-orbital injections. Compared with using transgenic animals, ICV viral injections enable rapid deployment of new GEVIs as soon they are developed. However, because neonatal ICV injection does not label neurons that develop at later stages, the cell population visualized differs from preparations where vectors are administered to adult mice. Second,

the components of our widefield imaging system (Fig. 3b) are commercially available, allowing end users to replicate our setup. The pre-processing pipeline we developed to correct for hemodynamic and motion artifacts is freely available (https://github.com/JaegerLab/JEDI-1P-Kv_widefield_imaging_preprocessing_pipeline)[71] and easily modifiable when optical recordings have different noise characteristics (Fig. 4). Of note, because green and red photons are differentially scattered and absorbed in tissue, breathing and hemodynamic

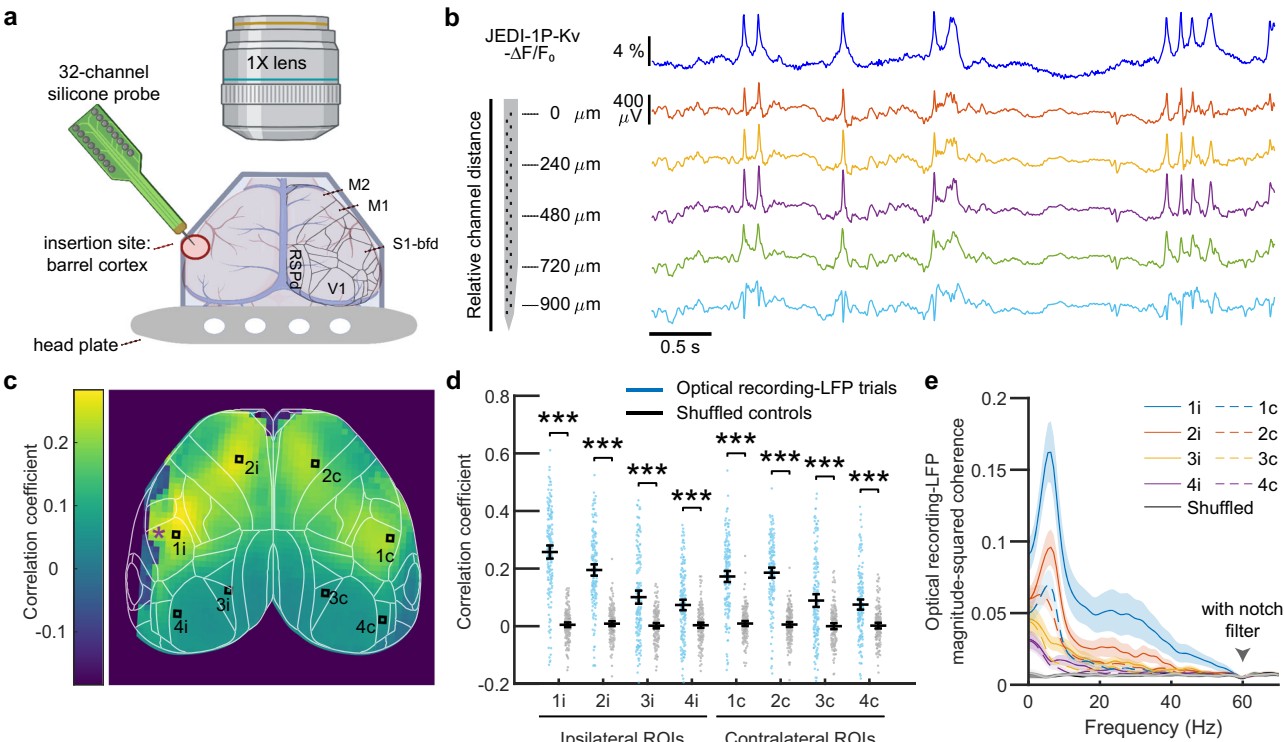

**Fig. 5 | JEDI-1P-Kv signals correlate with local field potentials (LFPs). a** Setup schematic for simultaneous voltage imaging and acute LFP recording in lightly anesthetized animals. Lines indicate the cortical map derived from the Allen Mouse Brain Common Coordinate Framework[92]. M1, primary motor cortex; M2, secondary motor cortex; S1-bfd, primary somatosensory (barrel) cortex; RSPd, retrosplenial cortex; V1, primary visual cortex. **b** JEDI-1P-Kv reports voltage transients also seen in LFP traces. Voltage imaging was conducted close (ROI 1i in (**c**)) to the LFP recording site (asterisk in (**c**)). LFPs were recorded from 32 channels, with inverted traces from 4 channels shown here (channels 1, 9, 17, 25, and 31). Depths are relative to the top channel. **c**, **d** LFPs are correlated with JEDI-1P-Kv signals from both ipsilateral (i) and contralateral (c) sides. **c** Map showing the mean correlation between the LFP recording and JEDI-1P-Kv signals from 2 × 2-pixel ROIs in a representative mouse. The purple asterisk indicates the LFP electrode insertion site. $n = 40$ trials. **d** LFP and JEDI-1P-Kv signals are correlated at each ROI ($p = 7.6 \times 10^{-42}$, $5.1 \times 10^{-39}$, $1.3 \times 10^{-9}$, $4.3 \times 10^{-9}$, $1.5 \times 10^{-34}$, $6.9 \times 10^{-40}$, $2.0 \times 10^{-8}$, $1.9 \times 10^{-11}$, two-sided Mann–Whitney $U$ test). For both groups at each ROI, $n = 172$ trials from 3 mice, with 40-84 trials/mouse. Shuffled control correlation coefficients are calculated from mismatched LFP recording and imaging trials. ROI locations are shown in (**c**). Dots are individual trials, the horizontal lines are the mean value, and the error bars are the 95% confidence intervals. **e** LFP and JEDI-1P-Kv signals are correlated below 60 Hz based on the magnitude-squared coherence. Note that the dip in the coherence at 60 Hz is expected due to the LFP notch filter at this frequency. Dark lines are the mean, and shaded areas are the 95% confidence intervals. Panel **a** was created using Biorender.com.

artifacts may modulate these wavelengths slightly differently, making it challenging to eliminate these artifacts entirely.

JEDI-1P has fast repolarization kinetics (i.e., off kinetics), and cortical pyramidal neurons spend less than 1% of time spiking on average[72]. Consequently, a significant fraction of the signals reported by JEDI-1P-Kv under our conditions are likely hyperpolarizations and subthreshold depolarizations—signals that calcium indicators cannot typically report. Our method enables long-term tracking of membrane voltage fluctuations and detection of physiologically important frequencies of up to 60 Hz. Because our approach could be applied to studying beta and gamma oscillatory coupling across the cortex, we anticipate it will generate valuable insights into the dynamic control of functional networks and the relationship between fast behavioral kinematics and their corresponding spatial neural response patterns.

Future work could combine supra-threshold measurements from calcium indicators with subthreshold recordings from JEDI-1P, either in separate animals, or by using a spectrally compatible calcium sensor[73,74]. Follow-up studies could also explore the relative contributions of different cortical cell types—including interneurons[75]—to distributed neural computations. While we demonstrated pan-cortical widefield voltage imaging, JEDI-1P could be used for reporting neural activity using other methods, including fiber photometry[58], miniaturized microscopes[70], confocal imaging[72], light sheet/SCAPE[76,77] and widefield imaging with cellular resolution[78]. We look forward to the broader neuroscience community leveraging JEDI-1P and our

optimized widefield imaging methods for understanding pan-cortical neural computations with cell type specificity and millisecond-timescale resolution.

## Methods

### High-throughput voltage indicator screening

**Reagents and buffers.** Cloning reagents include PrimeSTAR HS DNA Polymerase (R040A, TaKaRa), FastDigest NheI (FD0974, Thermo Fisher Scientific), FastDigest HindIII (FD0504, Thermo Fisher Scientific), FastDigest KpnI (FD0524, Thermo Fisher Scientific), FastDigest Bsp1407I (FD0933, Thermo Fisher Scientific), FastDigest Green Buffer (10X) (B72, Thermo Fisher Scientific), agarose (BP160-500, Fisher BioReagents), Tris-Acetate-EDTA (TAE) 50× Solution (BP1332-1, Fisher BioReagents), Lysogeny broth, Miller (BP1426-500, Fisher BioReagents), Ampicillin (BP1760-25, Fisher BioReagents), In-Fusion HD Cloning Kits (639650, TaKaRa), GeneJET Gel extraction kit (FERK0691, Fisher), PureLink Pro Quick96 Plasmid Purification Kit (K211004A, Thermo Fisher Scientific) and Mini Plus Plasmid DNA Extraction System (GF2002, Viogene).

Cell and neuron culture reagents include: high-glucose Dulbecco's Modified Eagle Medium (D1145, Sigma-Aldrich), fetal bovine serum (F2442, Sigma-Aldrich), glutamine (G7513, Sigma-Aldrich), Penicillin/Streptomycin (P4333, Sigma-Aldrich), Geneticin (G418) Sulfate (30-234-CR, Corning), phenol-free Neurobasal medium (12348017, Gibco), B-27 (17504044, Gibco), Glutamax (35050061, Gibco), 30-70 kD poly-

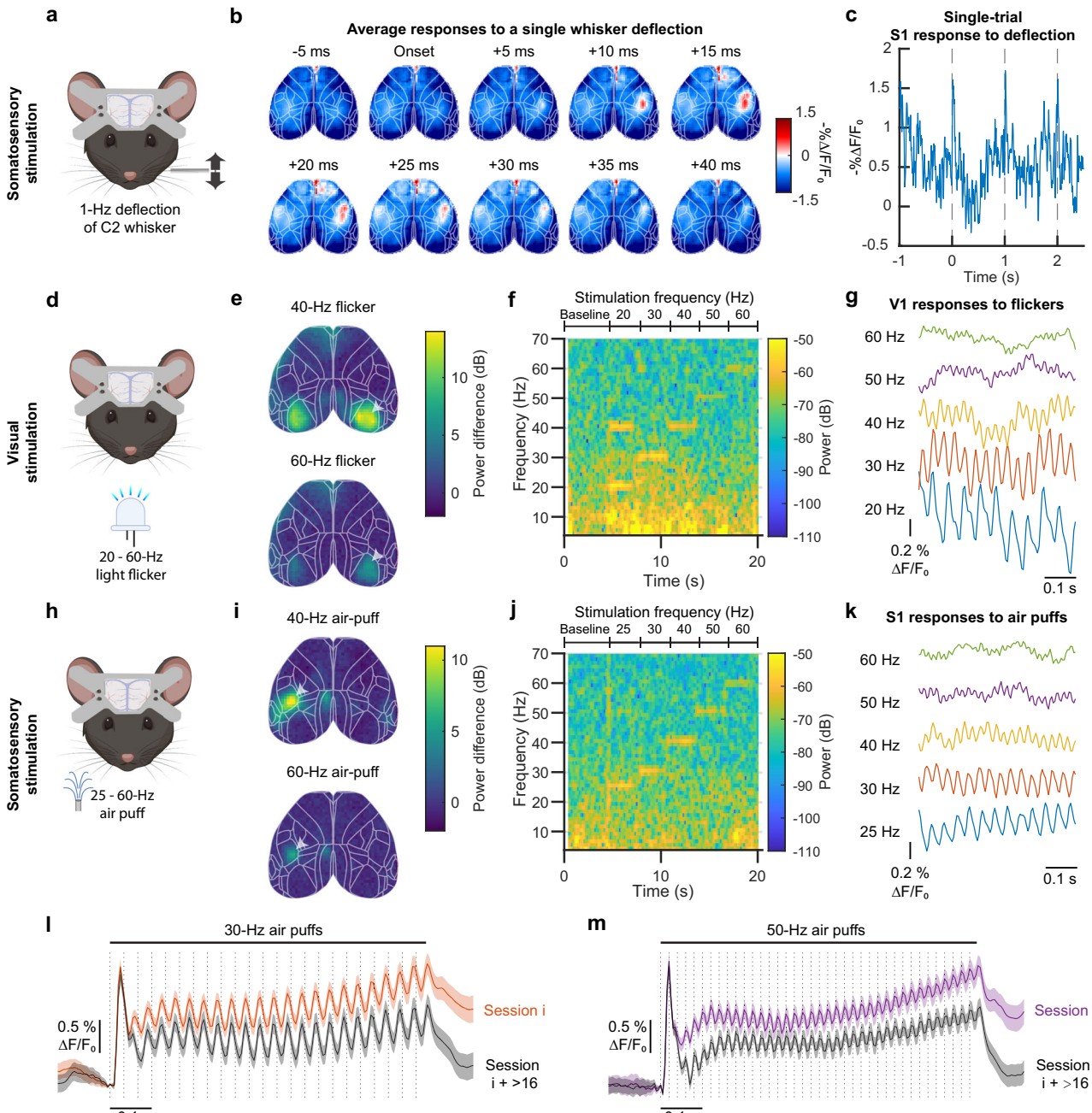

**Fig. 6 | JEDI-1P-Kv reports rapid voltage responses to high-frequency stimuli.**
**a–c** Simultaneous single whisker deflection and widefield voltage imaging in awake
mice. **a** Experimental schematics. **b** Contralateral barrel cortex shows rapid
response to a single whisker deflection. $n = 150$ deflections from an animal, representative of results with 5 mice. Responses of the other 4 mice are shown in Supplementary Fig. 11. **c** Example time series shows rapid responses to single whisker
deflections in a single trial. $n = 1$ mouse, representative of results with 5 mice.
**d–g** Simultaneous visual stimulation and widefield voltage imaging in awake mice.
**d** Experimental schematics. **e** 40-Hz-filtered (or 60-Hz) optical responses to 40-Hz
(or 60-Hz) visual stimulations are bilaterally localized in the visual cortex. The heat
map shows the mean power difference at 40 Hz (or 60 Hz) between stimulation and
baseline. $n = 4$ mice, 10 trials/mouse. **f** Single-trial imaging of JEDI-1P-Kv can report
local voltage responses at frequencies matching visual stimulations of up to 60 Hz.
Responses were acquired from a 2 × 2 pixel area in the visual cortex (arrow in (**e**)).
$n = 1$ mouse, representative of results with 4 mice. **g** Oscillations in the responses to
specific flicker frequencies are visible in averaged voltage signals up to 60 Hz. Data
from a 2 × 2 pixel area in the primary visual cortex. Data shown were acquired
1.5–2.0 s into a 3-s stimulus train. $n = 4$ mice, 10 trials/mouse. **h–k** Simultaneous

whisker stimulation and widefield voltage imaging in awake mice. **h** Experimental
schematic. Air-puff trains were applied to the right set of whiskers. **i** 40-Hz-filtered
(or 60-Hz) optical responses to 40-Hz (or 60-Hz) air puffs are localized at the
contralateral barrel cortex. The heat map shows the mean power difference at
40 Hz (or 60 Hz) between stimulation and baseline. $n = 5$ mice, 10 trials/mouse.
**j** Single-trial imaging of JEDI-1P-Kv can report local voltage responses at frequencies
matching air-puff stimulations of up to 60 Hz. Responses were acquired from a 2 ×
2-pixel area in the contralateral barrel cortex (arrow in (**i**)). $n = 1$ mouse, representative of results with 5 mice. **k** Same as (**g**) but for barrel cortex responses to
somatosensory stimuli. Data shown were acquired 1.5–2.0 s into a 3-s stimulus train.
$n = 5$ mice, 10 trials/mouse. **l, m** Responses to air puffs are stable over multiple days
and imaging sessions. Neural responses are delayed compared with the valve
opening triggers (dashed lines) due to the ~7-ms valve opening time. $n = 216$ trials
from 6 mice (36 trials/mouse). Gray traces were acquired 16–19 imaging sessions
after the orange and purple traces. The initial larger peak is consistent with electrophysiological data[93]. Dark traces are the mean, and shaded areas denote the 95%
CI. Panels (**a**, **d**, and **h**) were created using Biorender.com.

D-lysine (P7886, Sigma-Aldrich), 300 kD poly-D-lysine hydrobromide (P7405, Sigma-Aldrich), Trypsin-EDTA solution (T3924, Sigma-Aldrich) and phosphate-buffered saline (PBS, SH302560, HyClone, GE Healthcare).

Transfection reagents include JetPRIME (114-15, Polyplus Transfection), FuGENE HD transfection reagent (E2311, Promega), lipofectamine 2000 (11668019, Thermo Fisher Scientific), and Opti-MEM (31985-070, Gibco).

Reagents for solutions include: NaCl (S3014, Sigma-Aldrich), sucrose (S0389, Sigma-Aldrich), D-(+)-glucose (G8270, Sigma-Aldrich), HEPES (H3375, Sigma-Aldrich), KCl (P9541, Sigma-Aldrich), $MgSO_4$ (M2643, Sigma-Aldrich), K-gluconate (P1847, Sigma-Aldrich), EGTA (E3889, Sigma-Aldrich), $MgCl_2$ (M9272, Sigma-Aldrich), $CaCl_2$ (223506, Sigma-Aldrich), KOH (P250, ThermoFisher) and NaOH (S5881, Sigma-Aldrich).

Used for culturing HEK293-Kir2.1 cells, growth medium #1 contains high-glucose Dulbecco's Modified Eagle Medium supplemented with 10% fetal bovine serum (FBS), 2 mM glutamine, 100 unit/mL Penicillin, 100 μg/mL Streptomycin, and 750 μg/mL of the antibiotic G418 Sulfate (geneticin).

Used in transfection and HEK293A cell culturing, growth medium #2 contains high-glucose Dulbecco's Modified Eagle Medium supplemented with 5% fetal bovine serum (FBS), 2 mM glutamine, 100 unit/mL Penicillin and 100 μg/mL Streptomycin.

Used in high-throughput screening, whole-cell patch-clamp, spectra characterization, and neuron confocal imaging, the external solution is composed of 110 mM NaCl, 26 mM sucrose, 23 mM glucose, 5 mM HEPES, 5 mM KCl, 2.5 mM $CaCl_2$, 1.3 mM $MgSO_4$ and is titrated to pH 7.4 with NaOH.

**Mammalian cell lines.** HEK293A and HEK-Kir2.1 cell lines[43] were used in the study. HEK293A was purchased from Thermo Fisher Scientific (Cat #R705-07; RRID: CVCL_6910) and HEK293-Kir2.1 was a gift from Gui-Rong Li Lab.

**Plasmids construction and bacteria handling.** All plasmids were constructed through standard molecular biology methods and sequence verified by Sanger sequencing. All primers for cloning were synthesized through the Sigma-Aldrich standard DNA oligos service.

For vectors used for screening, GEVIs were fused to mCherry-CAAX at the C-terminus[46,79,80] via a self-skipping P2A linker[81] and were cloned in pcDNA3.1/Puro-CAG vector. ASAP1 and ASAP2s were subcloned from Addgene plasmids #52519 and #101274, respectively. ASAP3 was subcloned from a plasmid kindly provided by Dr. Michael Lin (Stanford). ASAP1-EGFP was cloned by replacing the circularly permuted GFP in ASAP1 (cpsfGFP-OPT) with EGFP (V2–K239[34]). ASAP1-dpOPT was cloned by replacing the cpsfGFP-OPT in ASAP1 with de-permuted cpsfGFP-OPT (dpsfGFP-OPT).

Polymerase chain reaction (PCR) was used to create single-site-saturation mutagenesis libraries. To achieve a uniform distribution of the amino acid, we mix a set of 4 forward primers at a molar ratio of 16:3:1:1, each containing a (degenerate) codon at the mutagenesis site: NNT (N = A, T, G or C), VAA (V = A, G, or C), ATG, and TGG. For each PCR reaction of 20 μL, the mixture of the forward primers and the reverse primer were each added to a final molar concentration of 0.5 μmol, together with 10-50 ng template plasmid and 10 μL 2×PrimeSTAR DNA polymerase premix. DNA was amplified using the following protocol: an initial denaturation step at 98 °C for 30 s; 35 amplification cycles of 98 °C for 10 s, 57 °C for 10 s, 72 °C for 1 min/kb of fragment length; a final extension step at 72 °C for 5 min. The backbone was linearized from pcDNA3.1/Puro-CAG vector with a ccdB-camR insert, using restriction enzymes NheI and HindIII. PCR products and linearized vector were both purified with gel electrophoresis and GeneJET Gel extraction and assembled via In-Fusion seamless cloning strategy per manufacturer's instructions. 1 uL In-Fusion reaction mix was transformed into commercial chemically competent *Escherichia coli* (XL10-Gold, Agilent). Colonies were manually picked into Lysogeny broth supplemented with 100 μg/mL ampicillin in a 96-well block the next day after transformation and incubated in a shaker (MaxQ 5000, Thermo Scientific) kept at 37 °C, and plasmids were purified 18 h after the inoculation with a 96-well plasmid purification kit (PureLink Pro, Thermo Fisher Scientific) following the manufacturer's instructions.

For vectors used for in vitro characterization in HEK293A cells, GEVIs were cloned in pcDNA3.1/Puro-CAG vectors with the same cloning strategy as constructing screening vectors except for no reference protein was attached, no degenerate codons were used, and the plasmids were purified with Viogene mini plus miniprep kit.

For vectors used for one-photon excitation and emission spectra characterization in HEK293-Kir2.1 cells, a control vector pcDNA3.1/Puro-CAG-EGFP-CAAX was constructed by fusing the CAAX membrane anchoring motif to the C-terminal of EGFP. The same cloning and DNA preparation strategies were used as the vectors used for in vitro characterization.

For vectors used for dissociated neuron expression, JEDI-1P was cloned into pAAV vector under the control of the neuron-specific hSyn promoter by replacing the ASAP2s sequence in pAAV-hSyn-ASAP2s (Addgene plasmid #101276). pAAV-hSyn vector was linearized with restriction enzymes KpnI and HindIII.

For vectors used for mice expression, we attached a Kv2.1 PRC motif[48] to the C-terminus of JEDI-1P through a triple GSS linker (GSSGSSGSS) to restrict the expression to soma and proximal dendrites. The double floxed inversed JEDI-1P-Kv2.1prc cassette was cloned into the pAAV vector under the control of EF1α promoter by replacing the hChR2(H134R)-EYFP sequence in pAAV-EF1α-double floxed-hChR2-EYFP-WPRE-HGHpA (Addgene plasmid #20298). pAAV-EF1a vector was linearized with restriction enzymes Bsp1407I and NheI.

**Mammalian cell culture and transfection.** For GEVI screening, we use a modified HEK293 cell line that stably expresses human Kir2.1 channel to maintain a resting membrane potential of around −77 mV[43]. Cells were cultured in Sanyo $CO_2$ incubators (MCO-18AIC(UV), Marshall Scientific) that are kept at 37 °C and supplemented with 5% $CO_2$. Growth medium #1 that contains 750 μg/mL of the antibiotic G418 Sulfate was used to maintain Kir2.1 expression in the cell line.

Transfections of HEK293-Kir2.1 cells for screening were performed in 96-well glass-bottomed plates (P96-1.5H-N, Cellvis). Plates were coated with 30-70 kD poly-D-lysine for 1 h at 37 °C to promote cell adhesion and washed twice with PBS before seeding the cells. HEK293-Kir2.1 cells were then seeded one day before forward transfection at 60–65% confluency, equivalent to 24,000–26,000 cells per well, or 1 h before reverse transfection at 70–80% confluency, equivalent to 28,000–32,000 cells per well. Growth medium #2 that is supplemented with 5% FBS and free from G418 was used for seeding the cells to improve cellular health while restricting the growth rate. Transfections were performed 36–48 h before screening with JetPRIME according to their protocol: for each well of the 96-well plate, 0.4 μL of JetPRIME reagent was first mixed in 10 μL of JetPRIME buffer, and then mixed with 130 ng of DNA prediluted in 10 μL JetPRIME buffer. The mixture was incubated for 8 min at room temperature before adding to the wells seeded with cells, and the medium in the 96-well plate was replaced with fresh growth medium #2 the next day to minimize the cytotoxicity of the transfection reagent.

**High-throughput GEVI screening platform.** To evaluate the performance of GEVIs in a high throughput way, we built an automated multimodal 96-well screening platform based on an inverted microscope (A1R-MP, Nikon Instruments). The excitation light was generated from an LED light engine (SpectraX, Lumencor) and directed to the microscope's epifluorescence illuminator via a liquid light guide. Cyan (470/24 nm, center wavelength/bandwidth) and green (550/15 nm)

light were delivered to the sample plan through a 20× 0.75 NA objective (CFI Plan Apochromat Lambda, Nikon Instruments) to excite the GFP-based GEVIs and mCherry, respectively. The emission light from the sample plane was split from the excitation light through a multi-band dichroic mirror (89000bs, Chroma) and filtered by a multi-band emission filter (89101m, Chroma) before being collected by a scientific complementary metal–oxide–semiconductor (sCMOS) camera (ORCA Flash 4.0 V2, C11440-22CU, Hamamatsu). A motorized extended travel stage capable (H139E1, Prior) was used to control the position of the field of view and to hold 96-well plates.

To support the automation of the system, data acquisition and output broads (PCI-6229 and PCI-6723, National Instruments) were connected to the microscope computer through a PXI Chassis (PXI-1033, National Instruments). The computer was equipped with 2 Intel Xeon E5-2630 v3 processors (total of 16 cores), 128 GB of DDR4 RAM, and four 2 TB SSDs in RAID 0 to facilitate high-speed imaging. JOBs scripts in NIS-Elements HC (version 4.60, Nikon Instruments) were used to control the microscope system (e.g., stage position), manage the optical configurations (e.g., excitations), initiate image acquisition, and trigger the stimulator.

A digital isolated high-power stimulator (4100, A-M System) was used to provide electric field stimulation. Electric pulses were passed to a pair of electrodes made from 0.5 mm wide platinum wires (99.95% pure, AA10286BU, Fisher Scientific). The two L-shaped electrodes had a horizontal length of 2 mm and were 3 mm apart (Fig. 1b), and they were secured on a 3D-printed polylactic acid holder. The holder was fixed to a motorized linear translation stage (MTS50-Z8, Thorlabs), controlled by Kinesis (Thorlabs) to move the electrodes in and out of individual wells. Two manual linear translation stages (411-05S, Newport) were used to fine-tune the electrodes' lateral position. During stimulation, the electrodes were submerged under the imaging solution, about 500 µm above the bottom.

**High-throughput GEVI screening under widefield one-photon illumination.** On the day of screening, medium in the 96 well plate was removed, and cells in each well were washed with 200 µL external solution twice before another 100 µL of external solution was added to each well as the imaging solution. To determine the brightness and the response amplitude of the sensors, 4 non-overlapping fields of view (FOVs) of 2048 × 200 (512 × 50 post 4 × 4 binning) pixels were imaged per well, allowing the sCMOS camera to record at a maximum speed of 987 frames per second (fps). The electric field stimulation (EFS) electrode was moved into the well after the motorized stage centered at the well and before the stage moved to the first selected FOV. The reference channel designated for the brightness of the mCherry was captured first under 555 nm illumination for 1 frame, with an irradiance of ~69 mW/mm² at the sample plane and an exposure time of 5 ms. Then, the target channel designated for capturing the responses from the green GEVIs was imaged at ~987 fps under 470 nm illumination for 4900 frames (~5 s), with an irradiance of ~36 mW/mm² at the sample plane and an averaged exposure time of 1.01 ms. Immediately before the camera started, a transistor-transistor logic (TTL) signal was sent to trigger the stimulator, which sent out a TTL signal to trigger the shutter of the light source and turn the light on. This allowed the camera to capture the onset of the excitation light, which eventually served as a reference point to locate the EFS events and temporally align the results from different FOVs. The stimulations took place 1 s after the light was turned on, starting from four monophasic square pulses with 1-ms width, 60-V amplitude, and a period of 500 ms. These monophasic pulses were followed by a 100-Hz train stimulation (10 monophasic square pulses with 2.5-ms width, 30-V amplitude, and a period of 10 ms), with a 1-s gap between the single pulses and the train stimulation. After the EFS on the 4 selected FOVs, the motorized stage centered in the well again, and the motorized linear translation stage raised the electrode outside of the well.

To determine the photostability of the sensors in each of the wells, another FOV of 2048 × 2044 (512 × 511 post 4 × 4 binning) pixels away from the previous FOVs was bleached for 2 min under 470 nm with an irradiance of ~36 mW/mm² at the sample plane, and the photobleaching video was recorded at a frame rate of 100 fps and an averaged exposure time of 10 ms. The excitation light was triggered the same way as described above (but no electrode in the well, so no EFS took place) to generate the light onset signal for aligning the results from different FOVs. Before each photobleaching video was taken, another single-frame reference channel for evaluating the mCherry intensity in the same FOV was captured with 555 nm LED excitation, with an irradiance of ~69 mW/mm² at the sample plane and an exposure time of 5 ms.

**High-throughput screening data analysis.** Time-lapse images collected from the screening were processed by custom routines in MATLAB (version r2021b, MathWorks). Recording files in Nikon NIS-Elements ND2 format were first parsed by the Bio-Formats MATLAB toolbox (version 6.3.1)[82]. For both channels of each FOV, saturated pixels, typically from over-expressing cells, were excluded and background levels were estimated using the intensity histogram and subtracted from images. To exclude pixels that do not represent cells, an initial mask was computed from the first frames of each channel using predefined thresholds. This mask was applied to the entire recording, and the overall change of fluorescence over time can be obtained by summing pixels selected by the mask frame by frame across all time points.

For electric field stimulation recordings, further pixel selection was necessary to mitigate pernicious influences on response amplitude due to overexpressing cells, bright extracellular fluorescent puncta, and intracellular aggregates. Based on the initial mask, a secondary mask with non-responsive pixels removed was generated for the green channel time series to obtain response amplitudes. To do this, we first estimated the photobleaching by fitting a three-term exponential curve on the overall fluorescence with time points during stimulation excluded. The time constants from the fitting were then used to estimate the trend for each pixel using the least-squares fitting. The trend of each pixel was then removed using division. After detrending, we calculated the correlation between the overall fluorescence and that of each pixel within the initial mask. These pixels were then ranked with decreasing correlation and grouped in batches of 200 pixels. The secondary mask was determined by accumulating groups of pixels from the highest correlation until the SNR reached a maximum. Fluorescence from the secondary mask was summed up and normalized by initial fluorescence for each FOV to quantify response amplitude.

For photobleaching recordings, the average fluorescence from pixels in the initial mask was normalized by the initial fluorescence. To quantify brightness, pixels of each channel from the initial mask were first averaged, and then the brightness was defined as the ratio of mean green fluorescence over red fluorescence.

## In vitro characterization of JEDI-1P
**Reagents and buffers.** Used for plating dissociated rat cortical neurons, the neuron plating medium contains Neurobasal medium supplemented with B-27, 2 mM Glutamax, 10% FBS, 100 unit/mL Penicillin, and 100 µg/mL Streptomycin.

Used for culturing dissociated rat cortical neurons, the neuron culturing medium contains phenol-free Neurobasal medium supplemented with B-27, 2 mM Glutamax, 100 unit/mL Penicillin, and 100 µg/mL Streptomycin.

NbActiv1 + Glutamate (NB1 + GLU, Transnetyx Inc.) is used for plating dissociated rat hippocampal neurons. This medium is part of the E18 Rat Dissociated Hippocampus kit (SKU SDEDHP, Transnetyx Inc.).

NbActiv4 medium (SKU NB4, Transnetyx Inc.) is used for culturing dissociated rat hippocampal neurons.

Used in whole-cell voltage-clamp, the internal solution #1 is composed of 115 mM K-gluconate, 10 mM HEPES, 10 mM EGTA, 10 mM glucose, 8 mM KCl, 5 mM $MgCl_2$, 1 mM $CaCl_2$ and is adjusted to pH 7.4 with KOH.

Used for whole-cell current-clamp in rat cortical neurons, the internal solution #2 contains 135 mM K-gluconate, 10 mM HEPES, 8 mM NaCl, 4 mM MgATP (A9187, Sigma-Aldrich), 0.4 mM Na-GTP (G8877, Sigma-Aldrich), 0.6 mM $MgCl_2$, 0.1 mM $CaCl_2$ and is adjusted to pH 7.25 with KOH.

Used for whole-cell current-clamp in rat cortical neurons, Tyrode's solution[23] contains 125 mM NaCl, 2 mM KCl, 3 mM $CaCl_2$, 1 mM $MgCl_2$, 10 mM HEPES, 30 mM glucose and is adjusted to pH 7.25 with NaOH.

Used for whole-cell voltage-clamp at different pH, aforementioned external solution is adjusted to pH 6.8–8 with HCl or NaOH.

**Mammalian cell culture and transfection.** For in vitro GEVI characterization with whole-cell voltage clamp, we used HEK293A cells (R70507, Thermo Fisher Scientific). The cell line was cultured under the same condition as HEK293-Kir2.1 cells, except that growth medium #2 was used with no G418 Sulfate supplement.

Transfections of HEK293A cells for patch clamp were performed in 24-well plates (P24-1.5H-N, Cellvis) on 12-mm cover glasses (#0, 633009, Carolina). Cover glasses were coated with 30–70 kD poly-D-lysine for 5–10 min at 37 °C to promote cell adhesion and washed twice with PBS before seeding the cells. HEK293A cells were seeded in growth medium #2 two days before patch-clamp at 30%–40% confluency, equivalent to 72,000–96,000 cells per well. Transfections were performed on the same day of seeding per manufacturer's instruction: For each well of the 24-well plate, 0.6 μL FuGENE transfection reagent that has pre-equilibrated at room temperature for 15 min was added to 200 ng DNA diluted in 13 μL Opti-MEM. The mixture was incubated for 7 min at room temperature before adding to the cells, and the medium was replaced with fresh growth medium #2 the next day.

**Whole-cell voltage clamp setup.** To evaluate the performance of GEVIs with whole-cell voltage clamp, we used GEVIs with no reference protein attached cloned in pcDNA3.1/Puro-CAG vectors and transfected into HEK293A cells as described in Cell culture and transfection. Electrophysiological recordings were done 36–48 h post-transfection at room temperature unless otherwise stated.

Pipettes for patch clamping were prepared freshly on the day of electrophysiology experiments from filamented glass capillaries (1B150F-4, World Precision Instruments) using a horizontal pipette puller (P1000, Sutter) to achieve a tip pipette resistance of 3–5 MΩ. The micropipettes filled with the internal solution #1 using MicroFil Flexible Needle (MF28G67-5, World Precision Instruments) were secured on a patch-clamp headstage (CV-7B, Molecular Devices) and positioned by a micromanipulator (SMX series, Sensapex).

The coverslip seeded with the transfected cells was placed in a custom glass-bottom chamber based on Chamlide EC (Live Cell Instrument) with the glass bottom made with a 24 × 24 mm #1 coverslip (89082-270, VWR). Cells were continuously perfused with the external solution at ~4 mL/min with a peristaltic pump (505DU, Watson Marlow). Whole-cell voltage clamp was achieved using a MultiClamp 700B amplifier (Molecular Devices). Patch clamp data was recorded with an Axon Digidata 1550B1 Low Noise system with HumSilencer (Molecular Devices). Cells were held at −70 mV at the baseline. A liquid junction potential of 11 mV was compensated in all command voltage waveforms. Recordings were considered satisfactory and were included in the final analysis only if the patched cell had an access resistance

(Ra) smaller than 12 MΩ and a membrane resistance (Rm) larger than 10 times Ra both before and after the recording.

All patch-clamp recordings were performed on the same inverted microscope used for GEVI screening. For characterization under one-photon illumination, cyan (470/24 nm, center wavelength/bandwidth) excitation light generated from the SpectraX light engine was delivered to the sample plane through a 40× 1.30 NA objective (CFI Plan Apochromat Lambda, Nikon Instruments) to excite the GEVIs at an irradiance of ~37.41 mW/mm² unless otherwise noted. For characterization under two-photon illumination, 920-nm excitation laser generated from a titanium:sapphire femtosecond laser (Chameleon Ultra II, Coherent) was directed to the sample plane by the resonant galvanometer scanners through the same 40× 1.30 NA objective (CFI Plan Apochromat Lambda, Nikon Instruments) to excite the GEVIs at a power of ~30 mW unless otherwise noted.

**Evaluating indicators' response amplitude under one-photon illumination.** To evaluate the sensors' response to action potentials (APs), we voltage-clamped the cells to follow a typical AP waveform that was recorded from a representative hippocampal neuron and modified to have an amplitude of 100 mV (from −70 mV baseline voltage) and a full width at half maximum (FWHM) of 2 ms to mimic the shape of layer 2/3 cortical neurons at room temperature[83]. We also modified a burst of APs recorded from the adult mouse somatosensory cortex L5 pyramidal neurons to mimic APs on top of subthreshold depolarizations. The AP burst waveform has a subthreshold depolarization of ~20 mV (from −70 mV baseline voltage) and APs of 60–90 mV amplitude (from −50 mV subthreshold voltage) and 3–4 ms FWHM. Cells were stimulated with 5 single 2-ms AP waveforms at 2 Hz, 5 single 4-ms AP waveforms at 2 Hz, 10 AP waveforms in train at 100 Hz, and the AP burst waveform, with around 3 s of interval holding at −70 mV between each two protocols. The voltage-clamped cell was centered in a FOV of 2048 × 200 (512 × 50 post 4 × 4 binning) pixels. The emission light from the cell was filtered by a multi-band emission filter (89101 m, Chroma) and collected by the ORCA Flash 4.0 V2 sCMOS camera at ~987 Hz with an average exposure time of 1.01 ms.

To evaluate the sensors' response to steady-state voltages, we applied 1-s step-voltage stimulations of −100, −80, −60, −40, −20, 0, 20, 30, and 50 mV to the voltage-clamped cells, with a 1.5 s gap holding at −70 mV between each two stimulations. The emission light was captured with the same optical configuration and camera settings as the AP capture described above.

Time series recordings from the camera were processed using a similar method for processing high-throughput electric field stimulation recordings (see above).

To evaluate the sensors' response to APs of different widths, we resampled the above-mentioned AP waveform recorded from hippocampal neuron to have a full-width at half-maximum of 0.25 ms, 0.5 ms, 1 ms, 2 ms, and 4 ms while maintaining the same shape and amplitude. Each cell was stimulated with 40 0.25-ms APs at 6.67 Hz, 20 0.5-ms AP at 6.67 Hz, 10 1-ms AP at 3.33 Hz, 10 2-ms AP at 3.33 Hz and 10 4-ms AP at 3.33 Hz, with 1-s gap holding at −70 mV between each two groups of stimulations. Fluorescent traces in response to APs of the same widths were averaged to calculate the response amplitude. Recordings were performed at 32–35 °C and maintained with a feedback-controlled inline heater system (inline heater SH-27B, controller TC-324C, cable with thermistor TA-29, Warner instruments).

To evaluate the sensors' response to high-frequency AP firings, we voltage-clamped the cells to follow 6 waveforms recorded from a Purkinje cell in the brain slice of a P57 male mouse at 34 °C. Each waveform is 1s long with 0–500 pA, 500-ms current injection occurred between 0.25 and 0.75 s, resulting in APs firings at 70 Hz (spontaneous, 0 pA injection), 138 Hz (100 pA), 202 Hz (200 pA), 242 Hz (300 pA), 278 Hz (400 pA) and 312 Hz (500 pA). The FWHMs of APs are between

0.28 and 0.40 ms. Recordings were conducted at 32–35 °C and maintained with a feedback-controlled inline heater system (inline heater SH-27B, controller TC-324C, cable with thermistor TA-29, Warner instruments). Spike detection and quantification of the detection accuracy were performed on the spikes between 0.25 and 0.75 s after the onset of each spike waveform using a customized MATLAB (version r2022a, MathWorks) routine similar to the algorithms used in VolPy[84], briefed as below:

The ground truth was defined as the temporal spike locations in the command voltage waveform. Spikes were identified with MATLAB function findpeaks, with a minimum peak prominence (MinPeakProminence) of 10 and a minimum peak distance (MinPeakDistance) of 0.001.

Spike detection in the fluorescence recording was carried out using the simple threshold method. The fluorescence trace averaged from the selected foreground pixel was filtered using a high pass filter of 60 Hz to remove the slow component in the recording (e.g., the baseline rising during current injections). The noise level σ was calculated as the standard deviation of a 500-ms baseline fluorescence before each AP waveform. Fluorescent spikes with amplitudes larger than 2 times the noise level were selected.

The accuracy of a sensor reporting the APs with fluorescence output was measured with a precision/recall framework. For each spike waveform, the time bin was calculated as the minimum distance between every two neighboring APs in the train. For each spike in the ground truth, if a spike is found in the fluorescence trace within ± time bin, we call this pair of spikes 'matched' and remove them from the ground truth/fluorescence spikes. After looping this matching progress from the first spike to the last spike in the ground truth, we define:

True Positive (TP) = number of matched pairs

False Positive (FP) = number of spikes identified in fluorescence trace but not in command voltage

False Negative (FN) = number of spikes identified in command voltage but not in the fluorescence trace

True Negative (TN) = 0

F1 score is then calculated following

Precision = TP / (TP + FP)

Recall = TP / (TP + FN)

F1 = 2 × Precision × Recall / (Precision + Recall)

To evaluate sensors' hysteresis effect, we first applied 1-s step voltage stimulations of −100, −80, −60, −40, −20, 0, 20, 30, and 50 mV to the voltage-clamped cells (i.e., forward ramp), followed with 50, 30, 20, 0, −20, −40, −60, −80, −100 mV 1-s voltage stimulations (i.e., reverse ramp) with a 1.5 s holding at −70 mV between each two stimulation and 3–4 s holding at −70 mV between forward and reverse ramps. To decipher the source of JEDI-2P's hysteresis effect under 1P, we subjected the voltage-clamped HEK293A cells holding at −70 mV to a 25–26 s 470-nm illumination before applying 1-s step voltage stimulations of −100, −80, −60, −40, −20, 0, 20, 30, and 50 mV. The same optical configuration and camera settings were used as in the AP/F-V curve characterization described above.

To evaluate JEDI-1P's response at different pH, voltage-clamped HEK293A cells were stimulated with 20 single 2-ms AP waveforms at 2 Hz, 10 AP waveforms in train at 100 Hz, and 1-s step-voltages of −100, −80, −60, −40, −20, 0, 20, 30, and 50 mV. The same optical configuration and camera settings were used as in the AP/F-V curve characterization described above.

To evaluate JEDI-1P's response under ultraviolet light excitation, voltage-clamped HEK293A cells were stimulated with 5 single 2-ms AP waveforms at 2 Hz, 10 AP waveforms in train at 100 Hz, and 1-s step-voltages of −100, −80, −60, −40, −20, 0, 20, 30, and 50 mV as described above. The same optical configurations and camera settings were used, except the excitation light was 395/25-nm, and the irradiance at the sample plane was 49.6 mW/mm².

**Evaluating indicators' response amplitude under two-photon illumination.** Two-photon characterizations of indicators' response amplitude were carried out using the same protocols as described previously[41] and briefed as below:

To evaluate the sensors' performance under 2PM, we used the same 2-ms FWHM AP and the 100 Hz AP waveforms as detailed in the 1PM characterization section. Cells were stimulated with 20 single 2-ms AP waveforms at 2 Hz, 10 AP waveforms in train at 100 Hz, and 1-s step-voltages of −100, −80, −60, −40, −20, 0, 20, 30, and 50 mV with 1.5 s of gap holding at −70 mV between each two voltage steps and 2–3 s of interval holding at −70 mV between each two protocols. The voltage-clamped cell was centered in a FOV of 512 × 32 pixels, and the emission light from the cell was split using a 560-nm dichroic mirror (348958, Chroma), filtered by a 525/50-nm bandpass filter (353716, Chroma) and detected by a gallium arsenide phosphide (GaAsP) photomultipliers tube (PMT) at 440 Hz. The detector PMT's gain was set to 20.

**Evaluating indicators' kinetics.** To evaluate the sensors' kinetics, we applied three 1-s 100-mV depolarization pulses from −70 to 30 mV. Between each two pulse, cells were held at −70 mV for 1.4 s. Electrophysiological recordings were conducted at 21–23 °C (room temperature) or 32–35 °C (closer to the 37 °C temperature of mice brains) using a feedback-controlled inline heater system (inline heater SH-27B, controller TC-324C, cable with thermistor TA-29, Warner instruments) to maintain the temperature in the perfusion chamber. The irradiance of the excitation light at the sample plane was tuned down to ~19.43 mW/mm² to minimize photobleaching, and a diaphragm was used to reduce the diameter of the excitation spot to ensure only the patch-clamped cell was illuminated.

To capture fluorescence changes at a higher temporal resolution than that of our camera, we used a multialkali photomultiplier tube (PMT, PMM02, Thorlabs) installed on one of the side ports of the microscope to collect the emitted photons from the GEVIs. A LabVIEW (version NXG 5.1, National Instruments) routine was used to control the PMT bias voltage and record the output voltage using the data acquisition and output boards. Data was collected at 80 kHz.

The output voltage from the PMT was analyzed by a custom routine written in MATLAB (version r2022b, MathWorks) to obtain the fluorescence signal for each cell. The raw data was first downsampled to 20 kHz. Then, the photobleaching correction was done by performing a three-term exponential fitting on the baseline (when the cell was held at −70 mV) and removing the trend from the entire signal using division. The corrected signal was cropped from 0.1-s before the estimated depolarization or repolarization onset to 1-s after the estimated depolarization or repolarization onset. The exact onset timing was fitted together with other coefficients with either single-exponential ($\mathbf{F(t)} = c + (k \times \exp((\mathbf{t} - t_O) \times \lambda)) \times (\mathbf{t} > t_O) + k \times (\mathbf{t} <= t_O))$ or dual-exponential $((\mathbf{F(t)} = c + (k \times \exp((\mathbf{t} - t_O) \times \lambda) + k_2 \times \exp((\mathbf{t} - t_O) \times \lambda_2)) \times (\mathbf{t} > t_O) + (k + k_2) \times (\mathbf{t} <= t_O))$ model where the $\mathbf{t}$ is the independent variable, $\mathbf{F}$ is the dependent variable, and the rest are the coefficients to be fitted. Among these coefficients, $c$ describes the mean plateau fluorescence, $k$ and $k_2$ describe the relative ratio of each exponential component, $\lambda$ and $\lambda_2$ describe the opposite reciprocal of time constants, and $t_O$ is an offset indicating the exact event onset timing.

**One-photon excitation and emission spectra characterization.** To characterize the one-photon excitation and emission spectra of JEDI-1P, we used the same vector used for in vitro GEVI characterization with whole-cell voltage clamp, i.e., pcDNA3.1/Puro-CAG-JEDI-1P, and a control vector pcDNA3.1/Puro-CAG-EGFP. HEK293-Kir2.1 cells were used to express JEDI-1P or EGFP to characterize the spectra of the indicator at a membrane voltage close to the resting membrane potential (−70 mV). Transfections were performed in 6-well plastic plates (3516, Corning) or 10-cm tissue culture dishes (353003, Corning), and HEK293-Kir2.1 cells were seeded one day before forward transfection at

60–65% confluency, equivalent to 720,000–780,000 cells per well or 5,280,000–5,720,000 cells per dish in growth medium #2. Transfections were performed 36–48 h before imaging using JetPRIME. The transfection medium was replaced 4 h after transfection with fresh growth medium #2 to minimize potential cytotoxicity from the transfection reagent.

On the day of the experiment, cells from 2 wells of the 6-well plate or 1/3 of a 10-cm dish transfected with the same construct were detached with trypsin, washed twice and diluted into the same external solution for screening and in vitro characterization adjusted to pH 6.8–7.6, and pooled into a single well of a glass-bottomed 96-well plate (P96-1.5H-N, Cellvis). Pooling the cells to a dense preparation was important to produce a strong signal that could be robustly detected by the plate reader. Untransfected cells were prepared with the same method to determine the background autofluorescence levels. A handheld automated cell counter (Scepter 3.0, Millipore) was used to plate a similar number of cells between conditions.

Spectra were determined using a plate reader (Cytation 5, BioTek) to quantify fluorescence from wells of the 96-well plates prepared above. Excitation spectra were acquired by scanning excitation wavelengths from 350 to 535 nm in increments of 1 nm and a bandwidth of 10 nm and collecting emission intensity at 560/10 nm. Emission spectra were acquired by exciting at 430/10 nm and measuring emitted photons from 460 to 650 nm in increments of 1 nm and a bandwidth of 10 nm. Individual scans of excitation and emission spectra were corrected for autofluorescence by subtracting the values from untransfected cells at each wavelength, and then normalized to their respective peaks. Spectra with excitation efficiency beyond ± 10% excitation efficiency at 350 or 535 nm were excluded due to the risk of improper autofluorescence correction. The final excitation and emission spectra were averaged from the normalized spectra of each individual scan. The peak excitation/emission wavelengths were averaged from the peak excitation/emission wavelengths of each individual scan.

**Photostability characterization under different 1P irradiances.** To characterize the photostability of the indicators, we used a laser diode light engine (LDI-WF, 15010-US) that can produce stronger illumination than our LED light source (SpectraX, Lumencor). HEK-Kir2.1 cells were seeded and transfected with pcDNA3.1/Puro-CAG-GEVI-P2A-mCherry-CAAX vectors as described in High-throughput GEVI screening under widefield one-photon illumination.

Two days after transfection, medium in the 96 well plate was removed, and cells in each well were washed with 200 μL external solution twice before another 100 μL of external solution was added to each well as the imaging solution. Photobleaching was conducted under 470 nm LDI excitation with an irradiance of 3.2–47 mW/mm² at the sample plane. Photobleaching videos were recorded at a frame rate of 100 fps and an average exposure time of 10 ms per frame under continuous illumination of 2 or 10 min. After each photobleaching video was taken, another single-frame reference channel for evaluating the mCherry intensity in the same FOV was captured with 555 nm LDI excitation, with an irradiance of ~72 mW/mm² at the sample plane and an exposure time of 5 ms.

**Photostability characterization under different 2P powers.** Comparison of indicators' photostability under two-photon was carried out using the same protocol as described previously[41] and briefed as below:

To characterize the photostability of the indicators under two-photon, we used the same titanium:sapphire femtosecond laser (Chameleon Ultra II, Coherent) as described in Evaluating indicators' response amplitude under two-photon illumination. HEK-Kir2.1 cells were seeded and transfected with pcDNA3.1/Puro-CAG-GEVI vectors as

described in High-throughput GEVI screening under widefield one-photon illumination.

Two days after transfection, medium in the 96 well plate was removed, and cells in each well were washed with 200 μL external solution twice, before another 100 μL of external solution was added to each well as the imaging solution. Excitation laser was generated from a titanium:sapphire femtosecond laser (Chameleon Ultra II, Coherent) and directed to the sample plane by the resonant galvanometer scanners through the same 20× 0.75 NA objective (CFI Plan Apochromat Lambda, Nikon Instruments). Photobleaching was conducted under 920 nm excitation with a power of 36–97 mW at the sample plane. For each well, four non-overlapping FOVs of 512 × 32 pixels were continuously imaged at a frame rate of 440 Hz for 5 min. The emission light from the cell was split using a 560-nm dichroic mirror (348958, Chroma), filtered by a 525/50-nm bandpass filter (353716, Chroma) and collected by a gallium arsenide phosphide (GaAsP) photomultipliers tube (PMT).

To calculate the half-lives of GEVIs, the videos were first background corrected and foreground-segmented using a predefined threshold as described in High-throughput screening data analysis. Averaged fluorescence from the foreground pixels was normalized to the fluorescence of the first frame of the video and then fitted with a three-term exponential function. Half-lives were calculated as the timing when the fitted curves reached 0.5 (half of the first-frame fluorescence).

**Brightness characterization under 2P.** Comparison of indicators' brightness under two-photon was performed using the same protocol as described previously[41] and briefed as below:

To characterize the two-photon brightness of the indicators, we used the same titanium:sapphire femtosecond laser (Chameleon Ultra II, Coherent) as described in Evaluating indicators' response amplitude under two-photon illumination. HEK-Kir2.1 cells were seeded and transfected with pcDNA3.1/Puro-CAG-GEVI-GSS-cyOFP1 vectors as described in High-throughput GEVI screening under widefield one-photon illumination.

Two days after transfection, medium in the 96 well plate was removed, and cells in each well were washed with 200 μL external solution twice before another 100 μL of external solution was added to each well as the imaging solution. An excitation laser of 920-nm was generated from a titanium:sapphire femtosecond laser (Chameleon Ultra II, Coherent) and directed to the sample plane by the resonant galvanometer scanners through the same 20× 0.75 NA objective (CFI Plan Apochromat Lambda, Nikon Instruments). The emission light from the cell was split using a 560-nm dichroic mirror (348958, Chroma), filtered by a 525/50-nm (353716, Chroma) and a 605/70-nm (350069, Chroma) bandpass filter for the green and red channel respectively, and collected by gallium arsenide phosphide (GaAsP) photomultipliers tubes (PMTs). For each well, four non-overlapping FOVs of 512 × 32 pixels were first scanned for 50 frames (0.22 s) to collect the red-emission light from the reference protein, and then field stimulated with simultaneous imaging at 440 Hz for 4000 frames (8.4 s) to collect the responses from the green-emitting indicators. The field stimulation protocol contains 20 monophasic square pulses at 3.33 Hz (1-ms width, 60-V amplitude, and a period of 300 ms) followed by a 100-Hz train stimulation (10 monophasic square pulses with 2.5-ms width, 30-V amplitude, and a period of 10 ms).

To quantify the brightness, videos collected in the field stimulation were analyzed as described in *High-throughput screening data analysis*, except that the initial mask is generated from the first 20 frames in the red channel. For each FOV, a G/R score was computed using the secondary mask (marking all responding pixels) as the mean fluorescence of the first 20 frames of the green channel divided by the mean fluorescence of the first 20 frames of the red channel. The

brightness of each well is calculated as the mean G/R score of the four FOVs, weighted by the pixel number in the secondary mask.

**Confocal imaging of GEVI in dissociated neurons.** To determine the expression and trafficking of GEVIs in neurons, we used rat cortical neurons. Primary rat cortical neurons were isolated from day 18 Long-Evans rat embryos. Cortices were dissected, dissociated with papain (Worthington Biochemical Corporation), washed with trypsin inhibitor (Sigma), and seeded at 500,000 cells/mL in 500 μL neuron plating medium per well of a 24-well glass bottom plate (P24-1.5H-N, Cellvis). The plate was pre-coated overnight with 300 kD poly-D-lysine hydrobromide and washed twice with PBS before seeding. The plating day was considered as day in vitro (DIV) 0. The next day, 90% of the media was replaced with the neuron culturing medium. Half of the media was henceforth replaced with fresh neuron culturing medium every 3-4 days. All media were pre-equilibrated for at least 24 h at 37 °C in air with 5% $CO_2$ before usage. Neurons were kept in Sanyo incubators (MCO-18AIC(UV), Marshall scientific) that are kept at 37 °C and supplemented with 5% $CO_2$ after plating, and transfected at DIV 9 using 1 μL lipofectamine 2000 and 800 ng total DNA, including 200 ng pAAV-hSyn-JEDI-1P and 600 ng pNCS bacterial expression vector as buffer/filler DNA.

Six days after transfection (DIV 15), the attached neurons were washed twice with the external solution before adding a final 500 μL per well as the imaging solution. Laser-scanning confocal images were obtained using a high-speed confocal microscope (LSM880 with Airyscan, Zeiss) driven by the Zen software (version 2.3 SP1 FP3 black edition, Zeiss). The microscope was equipped with a 40× 1.1 NA water immersion objective (LD C-Apochromat Korr M27, Zeiss), a 488-nm argon laser (LGK7812, Lasos) set to 20% power (~200 μW) and a per-pixel dwell time of 2.35 μs. Emission light was filtered using a multipass beamsplitter (MBS 488/561/633, Zeiss) and acquired with a 32-channel GaAsP detector (Airyscan, Zeiss) with a detector gain of 800, and a 2.15-Airy unit pinhole size. Images were acquired at a resolution of 0.05 μm/pixel and an X-Y dimension of 3544 × 3544 pixels. Z stacks were stepped at 0.22 μm between images. Figure 2k corresponds to a maximum intensity projection from Z-stack with 25 images. Airyscan processing was applied to the images via Zen software (version 2.3, blue edition, Zeiss) to increase the resolution.

**Evaluating indicators' response amplitude in dissociated neurons.** To evaluate the indicators' performance in neurons in vitro, we used dissociated rat cortical neurons and hippocampal neurons.

To prepare the coverslip for seeding neurons, 12-mm cover glass (#0, 633009, Carolina) was etched with nitric acid (438073, Sigma-Aldrich) overnight and washed with Milli-Q water daily for 4 weeks. Etching and washing were performed in a capped glass conical flask shaking at 150 rpm on an orbital shaker (88882005, Thermo Scientific). The coverslips were then washed with ethanol, dried on filter papers, and autoclaved before use.

E18 rat cortical neurons were prepared and cultured as mentioned in Confocal imaging of GEVI in dissociated neurons, except the cells were seeded on the cover glass rather than in glass-bottom 24-well plates. Transfection was conducted on DIV 7 using 1 μL lipofectamine 2000 and 800 ng total DNA, including 150 ng pAAV-hSyn-JEDI-1P-Kv and 650 ng pNCS bacterial expression vector as buffer/filler DNA. Four days after transfection (DIV 12), cortical neurons were current-clamped in Tyrode's solution[23] at room temperature.

E18 rat hippocampal neurons were purchased from Transnetyx tissue by BrainBits (SKU SDEDHP, Transnetyx, Inc.) and came in Hibernate® EB complete Media as single-cell suspensions. Prior to seeding the hippocampal neurons, the coverslips were coated with Neuron Coating Solution (027-05, Sigma-Aldrich) at 37 °C overnight and washed three times with PBS. Neurons were seeded at 80,000–100,000 cells/mL in 500 μL glutamate-supplemented

NbActiv1 medium per well of the 24-well plate following the manufacturer's protocol. The plating day was considered as DIV 0. Half of the media was henceforth replaced with fresh, pre-equilibrated NbActiv4 medium every 3–4 days. Transfection was conducted on DIV 12–14 using 1 μL lipofectamine 2000 and 800 ng total DNA, including 80–100 ng pAAV-hSyn-JEDI-1P-Kv or pAAV-hSyn-ASAP3-Kv and 680–700 ng pNCS bacterial expression vector as buffer/filler DNA. Two days after transfection (DIV 14–16), hippocampal neurons were current-clamped in the external solution at 32–35 °C maintained with a feedback-controlled inline heater system (inline heater SH-27B, controller TC-324C, cable with thermistor TA-29, Warner instruments).

For current clamp in both types of neurons, glass pipettes were prepared with the same protocol as those prepared for voltage clamp in HEK 293 A cells, detailed in Whole-cell voltage clamp setup.

For current clamp in cortical neurons, glass pipettes were filled with internal solution #2. Current-clamp recordings were achieved using the same MultiClamp 700B amplifier (Molecular Devices) and Axon Digidata 1550B1 Low Noise system with HumSilencer (Molecular Devices) as used for whole-cell voltage clamp. After break-in, cells were held at 0 pA at the baseline. Action potentials or other membrane voltage changes were evoked with 1-s, 10–100 pA current injection and recorded through Clampex (Molecular Devices). A liquid junction potential of 16.1 mV was compensated post hoc in the recorded voltage waveforms. Concurrent with current-clamping, voltage imaging was conducted at 987 Hz under irradiance of ~190 mW/mm² with 470 nm excitation from LDI (LDI-WF 15010-US, 89 North). The rest of the optical setups were the same as described in Whole-cell voltage clamp setup.

For current clamp in hippocampal neurons, glass pipettes were filled with internal solution #1 supplemented with 4 mM MgATP and 0.4 mM Na-GTP. The current clamp was achieved with the same setup as used for cortical neurons, except that action potentials were evoked with 20-ms, 50–400 pA current injection. A liquid junction potential of 11 mV was compensated post hoc in the recorded voltage waveforms. Concurrent with current-clamping, voltage imaging was conducted at 987 Hz under irradiance of ~37–77 mW/mm² with 470 nm excitation from SpectraX (Lumencor). The rest of the optical setups were the same as described in Whole-cell voltage clamp setup. Only neurons with resting membrane potential lower than −45 mV were included in the data analysis. Among qualified neurons, only APs with larger than 45 mV peak amplitude (from resting membrane potential) and a width of 1–5 ms at 75% of the peak height were considered qualified. The final statistics was drawn from an average of 4–19 APs per neuron.

**Widefield imaging of JEDI-1P in mouse**
**Animals.** 14 male and 15 female *EMX1*-Cre mice were used for experiments. Animals were provided with *ad libitum* food and water unless specified otherwise. Mice were maintained at 72 °F (22 °C) with the humidity ranges from 30–70%. For imaging experiments, mice that received imaging window and head plate implantation surgery were singly housed with enrichment under a reverse cycle (12 h light, 12 h dark). Imaging experiments were performed during the dark cycle. Mice were placed on continuous water restriction for awake imaging. Emory University Institutional Animal Care and Use Committee approved all animal work performed in this study.

**Neonatal intracerebroventricular injection.** To achieve brain-wide expression of JEDI-1P and reference fluorescence, we used intracerebroventricular injection to deliver viral vectors[61]. *EMX1*-Cre mice were used to set up mating pairs. Breeder cages were monitored twice daily approaching the expected delivery day. Newborn pups were carefully retrieved from the breeder cage after they showed visible milk spot. The pups were then placed on a heating pad kept at 37 °C during the preparation of viral vectors. AAV.PHP.eB-EF1a-DIO-JEDI-1P-Kv2.1-WPRE (2–4 × 10¹² vg/mL) and AAV9-hSyn-mCherry (1.3 × 10¹³ vg/mL) were

mixed with a particle ratio of 1:1.3. For mice that received tdTomato instead of mCherry, the viral solution consisted of AAV9-EF1a-DIO-JEDI-1P-Kv2.1-WPRE ($1.9 \times 10^{13}$ vg/mL) and AAV.PHP.eB-CAG-tdTomato ($3.1 \times 10^{13}$ vg/mL) with a particle ratio of 36:1. Both combinations achieved similarly robust expression of JEDI-1P and the reference fluorescence and enabled imaging of neural signals at the cortical level. For injection in each pup, a total of 5 μL mixture was loaded into a 10-μL Nanofil syringe (World Precision Instrument) with a 34 G beveled needle (NF34BV-2, World Precision Instrument). After identifying the injection site, 2/5 of the distance between the lambda suture to each eye, the syringe was carefully inserted at the target to a depth of 3 mm. 2 μL of viral vectors were slowly dispensed on each hemisphere. After all the injections were completed, the pups were returned to the breeder cage.

**Histology.** Mice were injected with 0.1 mL of Euthasol (51311-0050-01, Virbac) prior to transcardial perfusion. The animals were then perfused with 0.1 M phosphate buffer (PB) followed by 4% paraformaldehyde (PFA) PB solution. The harvested brain tissue was left in 4% PFA PB solution for 24 h before being transferred to 15% sucrose PB and then 30% sucrose PB. The brain tissue was sectioned with a microtome (HM430, Epredia) at 50 μm thickness and imaged with a fluorescence microscope (BZ-X, KEYENCE).

**Surgery of imaging window and head-post implant.** The imaging window for in vivo widefield imaging was adapted from the clear-skull method[62,63]. 3% isoflurane was used for induction of anesthesia, and mice were kept under 2% isoflurane during the surgery. After fur removal with Nair (I0041395, Church & Dwight), the scalp was cleaned with alternating 70% ethanol and povidone-iodine pads. The scalp was removed to expose about $9 \times 9$ mm of the skull. The skull was gently scraped with a scalpel to remove fascia and dried with cotton swabs. A thin layer of Opti-bond Universal (36519, Kerr) was then applied to the dried surface. After Opti-bond was cured with ultraviolet light, a layer of dental cement mixture was applied before placing the custom cut coverslip (12-545-88, Fisherbrand Superslip) in place. The coverslip was cut with a diamond wedge scribe (S90W, Thorlabs) to a hexagon shape that was fitted to the inside of a head-plate (Fig. 3c). The dental cement for creating clear imaging window was mixed with 1 scoop of C&B Metabond L-powder and 5 drops (S399, Parkell Products Inc.) of C&B Metabond Quick Base (S398, Parkell Products Inc.). Head plate for head-fixation was also secured with the dental cement (2 scoops of L-powder and 6 drops of Quick Base) after the imaging window was cured in place.

**Continuous water restriction.** For awake imaging sessions, mice were placed under water restriction at least 2 days prior to the experiments. On day one of the water restriction, the ad libitum water was removed. The weight of the animal at the time of water removal was recorded as the baseline weight. The animal was weighed daily and was given a minimum of 40 ml/kg water each day (according to its baseline weight) to maintain a weight above 80% of the baseline. Continuous water restriction lasted no longer than 2 months.

**In vivo wide-field imaging.** A CMOS high-speed imaging system (MiCAM ULTIMA. SciMedia Ltd.) with dual cameras was used for in vivo widefield imaging at 200 Hz (Fig. 3b). JEDI-1P-Kv and mCherry were excited with a single blue LED (466/40 nm, center wavelength/bandwidth, FF01-466/40, Semrock) through the imaging window. The power of excitation light was 0.05 mW/mm² at the plane of the imaging window for all experiments. Emission light from JEDI-1P-Kv and mCherry was conditioned by 525/50 nm (Semrock: FF03-525/50-50) and 650/60 nm (FF01-650/60-50, Semrock) emission filters,

respectively. For imaging of mice with JEDI-1P-Kv and tdTomato, the emission filter for JEDI-1P-Kv was replaced with 515/30 nm (FF03-515/30-50, Semrock) to avoid emission from tdTomato above 530 nm. A dichroic beam splitter at 495 nm (FF495-Di03-50×70, Semrock) was used to reflect excitation light, and another dichroic at 580 nm (FF580-FDi01-50 × 70, Semrock) was used to split the emission light into two cameras. The spatial resolution of each camera was $100 \times 100$ pixels, and the field of view was $10 \times 10$ mm in size.

**Electrocardiography recording.** To qualitatively verify that fast hemodynamic signals are present in the imaging signals, we recorded electrocardiography (EKG) simultaneously during imaging. The fur on the animals' chests was shaved on the day of recording. Mice were lightly anesthetized with 1% isoflurane during EKG recording. A circular-shaped electrode with Spectra 360 electrode gel (12-08, Parker Labs) was secured onto the shaved chest area. The EKG signals were amplified through Extracellular Amplifier (EXT-02F, NPI Electronic GmbH) and recorded simultaneously during imaging.

**Widefield imaging data processing.** We developed a pipeline that uses reference fluorescence to subtract non-voltage signals from JEDI-1P-Kv signal. We implemented the data processing algorithms in MATLAB (version R2021b). The background fluorescence, estimated by measuring autofluorescence from control mice ($n = 4$ mice) under the same imaging condition, was first subtracted from both JEDI-1P-Kv channel and the reference channel. Then the photobleaching was detrended from the background-subtracted fluorescence JEDI-1P-Kv trace. The fast photobleaching was calculated empirically from averaging 5536 JEDI-1P-Kv traces (20.48 s long, $n = 6$ mice, 692 trials, 8 traces/trial), which were digitally filtered with a lowpass filter (lowpass function) at 0.5 Hz. The mean fast photobleaching was then fitted to a two-term exponential model. We also employed the same process to estimate the photobleaching from mCherry but found no apparent photobleaching in the mCherry channel. After background fluorescence subtraction and photobleaching removal, the change of fluorescence was calculated using $\Delta F/F_0 = (F - F_0)/F_0$. Baseline $F_0$ was determined from the mean fluorescence from time interval $T = 2$ s and $T = 2.5$ s. Reference $\Delta F/F_0$ was digitally filtered using lowpass and highpass filters (lowpass and highpass functions) at three ranges: [10, 30] Hz, [1, 10] Hz, and below 1 Hz unless specified otherwise. Frequency ranges [4, 20] Hz, [1, 4] Hz, and below 1 Hz were used for anesthetized recording due to slower heartbeat. For awake simultaneous LFP and imaging recording, filters at [14, 30] Hz, [10, 14] Hz, [5, 10] Hz, [1, 5] Hz, and below 1 Hz were applied to the reference channel. JEDI-1P-Kv $\Delta F/F_0$ was filtered using lowpass filter at 70 Hz. In Step 1, reference $\Delta F/F_0$ $r_1$ was scaled and subtracted from JEDI-1P-Kv $\Delta F/F_0$ $g_1$ and formed $g_2$ using Ordinary Least Squares (OLS) regression. In Step 2, $r_2$ was scaled to and subtracted from $g_2$ to form $g_3$. The same regression process scaled and removed $r_3$ from $g_3$. The same process can be repeated until shared noises at different frequency ranges have been removed. To compare JEDI-1P-Kv signal before and after the step-by-step regression, we calculated the mean power spectral density of the signal before and after the regression step ($n = 25$ trials) with fast Fourier transform (FFT) using MATLAB Spectrum Analyzer. The same power spectral analysis was used to compare the frequency distribution of JEDI-1P-Kv and reference channels.

**Simultaneous widefield imaging and acute local field potential recording in lightly anesthetized and awake mice.** For simultaneous widefield imaging and acute local field potential (LFP) recording, mice first received the imaging window and head-post implant and were allowed at least a week to recover.

The animals were anesthetized with 2% isoflurane and kept on a 37 °C heating pad while a small craniotomy (1 × 2 mm) was carefully drilled above the barrel cortex at the edge of the imaging window (Fig. 5a). After the skull was carefully lifted and the cortex was exposed, a thin layer of Dura-Gel (Cambridge Neurotech) was applied over the dura to prevent the exposed area from drying. Another small craniotomy was created above the cerebellum or above olfactory bulb in order to place a stainless-steel grounding wire. The anesthetized recording immediately followed the craniotomy procedure. For the awake recording, animals recovered from isoflurane for at least 4 hours before the recording.

The isoflurane was reduced to 1% during the lightly anesthetized recording, and the animals were kept on the 37 °C heating pad and head-fixed throughout the procedure. A 32-channel silicone probe (ASSY-37H8B, Cambridge Neurotech) was inserted into the craniotomy above the barrel cortex at a 45-degree angle of 1.0–1.2 mm deep. The LFP data were acquired through RHD2000 USB Interface Board (Intan Technologies, LLC) at 20 kS/s, and the imaging data were acquired through MiCAM ULTIMA at 200 Hz. Each trial with simultaneous imaging and LFP recording was 20.48 s long. The recording setup was the same for the awake state aside from the lack of anesthesia. Awake mice received air-puff stimulation on one side of their whiskers in a subset of trials. A trial with air-puff stimulation consisted of 4.5 s baseline and five 3-s long stimulation periods with 25, 30, 40, 50, and 60 Hz air-puff trains.

**Data analysis of widefield simultaneous imaging and acute LFP recording.** To show the relationship between voltage imaging and LFP, we calculated the correlation coefficient between the two types of data ($n = 3$ mice for anesthetized recording and $n = 2$ mice for awake recording). Trials with high-amplitude noise in LFP recording caused by respiration or motion were excluded from the analysis. First, we identified the channel of the neural probe that best correlated with imaging signals. Inverted LFP signals were down-sampled to 200 Hz to match the imaging acquisition rate and lowpass filtered at 70 Hz. LFP signals also went through a notch filter at 60 Hz for all analyses unless otherwise noted. To show the coherence at 60 Hz during 60 Hz air-puff stimulation in awake mice, the notch filter was not applied to LFP. We sampled 8 ROIs from each mouse and calculated the correlation coefficient between the regressed JEDI-1P-Kv trace at each ROI and the matching LFP signals from all 32 channels using MATLAB function corrcoef. The correlation coefficient between the same JEDI-1P-Kv signal and all electrode channels varied based on the depth of electrode channels in the cortex. For each set of matching trials with the same probe insertion depth, the modal electrode channel with maximum correlation coefficient among all 8 ROIs was identified. The modal channel number was used for the following analyses.

To understand the spatial relationship between LFP signal and cortex-wide imaging signals, we calculated the correlation coefficient between the LFP and JEDI-1P-Kv signal at each pixel. Within each animal, we averaged pixel-wide correlation coefficient across all trials to form the two-dimensional heat map that shows the correlation coefficient of the entire cortex (Fig. 5d and Supplementary Fig. 10c, f). To explore the relationship between LFP and imaging signal at selected ROIs, we averaged the correlation coefficient at each ROI from all animals and compared it with shuffled controls. The shuffled controls were generated by randomly mismatching LFP and imaging trials from all mice of the same experimental setting (172 trials from 3 mice for anesthetized recording, 112 trials from 2 mice for awake recording without stimulation, and 114 trials from 2 mice for awake recording with stimulation). A Mann-Whitney $U$ test was used to compare the correlation coefficient at selected ROIs from matched trials with that from shuffled controls. With the same imaging and LFP dataset, magnitude-squared coherence was calculated using MATLAB function mscohere to show the similarity of the two types of data in the frequency domain of interest.

**In vivo imaging with sensory stimulation.** We applied single whisker deflection, air-puff, or light flicker stimulations to mice during widefield imaging to test JEDI-1P's ability to follow fast frequency responses in the cortex. The animals were head-fixed during the imaging. All trials were 20.48 s long. For single whisker deflections, the left C2 whisker was threaded into a glass pipette connected to a piezo strip. The surrounding whiskers were trimmed. The C2 whisker was deflected at 1 Hz frequency with 0.5% duty cycle for 15 s following the baseline. For air-puff stimulation, the air was delivered via a metal tube placed vertically under the right-side whiskers and controlled through a miniature solenoid valve (LHDA1233215H, Lee Company). For light flicker experiments, a white LED was placed 20 mm away from the animals' eyes with an illuminance of 400 Lux[85]. Each imaging trial started with a 5 s baseline with no stimuli and was followed by stimulation at 25 (or 20 for light flicker), 30, 40, 50, and 60 Hz with 50% duty cycle that lasted 3 s at each frequency.

For experiments showing stimulation response to sensory stimuli during long-term imaging, mice received a 0.75 s air-puff stimulation at either 30 Hz or 50 Hz with 50% duty cycle. The imaging was 10.24 s per trial.

**Data analysis of in vivo imaging with sensory stimulation.** To demonstrate the fast response to single whisker deflection of JEDI-1P in vivo, we plotted the averaged spatial maps $\Delta F/F_0$ of individual animals ($n = 150$ deflections/animal) and the average of all mice ($n = 5$ mice).

To show the response at high-frequency range during air-puff stimulation or light flicker, the spatial maps of power at 40 Hz and 60 Hz were averaged across mice. The cortical images were shifted to a standard cortical map by aligning the superior sagittal sinus to the midline as well as S1 and V1 activation to the corresponding regions. The co-registered maps enabled averaging across sessions and animals. Power at 40 Hz or 60 Hz was calculated for baseline and corresponding stimulation periods using MATLAB function spectrogram. The power difference between stimulation and baseline was calculated for each 2×2-pixel on a trial-by-trial basis. For each animal, power difference from 10 trials was averaged. The power difference at 40 Hz and 60 Hz in Fig. 6e, i were averaged from 4 mice and 5 mice, respectively. To highlight that the response to different frequencies of stimuli was observable in a single trial, spectrogram of JEDI-1P response was calculated from the signal from a region of interest (Fig. 6f, j) using MATLAB function spectrogram.

**Statistical analysis**
For each of the comparisons we made in the manuscript, statistical details were specified either in the main text or along with the figure captions, including (1) the statistical test used, (2) sample sizes $n$, (3) what $n$ represents, (3) the definition of the center (i.e., mean or median) and (5) the definition of the error bars. Detailed values for each graph are provided in the Statistical Information spreadsheet provided as Supplementary Information. A comparison is defined to be statistically significant if the $p$-value was less than 0.05, unless otherwise stated.

For in vitro experiments, we performed two-tailed $t$ tests when comparing the means between two groups and ANOVA when comparing the means among more than two groups. Prior to $t$ test, one-way and two-way ANOVA, we first conducted $F$ test, Brown–Forsythe test and Spearman's test, respectively, to test for equal variances between groups. When the variances were statistically different, Welch's correction was applied. For two-way ANOVA, reported p-scores are associated with the categorical factors from which the comparison was made unless otherwise noted. Following one-way ordinary ANOVA,

Welch's ANOVA and two-way ANOVA, we conducted Tukey's Hones Significant Difference (HSD), Dunnett's T3 and Holm-Šídák post hoc test, respectively, for multiple comparisons. Because normality tests have little power when the sample size is small[86,87], we did not perform the normality check and assumed normal distribution when appropriate.

For linear regression we did for in vitro experiments, statistical details were specified either in the main text or along with the figure captions, including (1) how regression was performed, (2) sample sizes $n$, (3) what $n$ represents, Result was reported as $r$-squared effective size (Pearson's $r^2$). Statistical analyses were performed with GraphPad Prism 9 or Wolfram Mathematica 13.

For in vivo experiments, we performed Mann–Whitney $U$ test when comparing the experimental group and the control group. Sample size $n$ and $p$-value for each test were specified in the caption. Functions sigstar() and stdshade() were adapted for plotting significance stars and 95% CI, respectively[88,89]. Function notBoxPlot() was adapted to plot mean, 95% CI, and standard deviation[90].

## Statistics and reproducibility
The expression of GEVI-P2A-mCherry-CAAX constructs in HEK-Kir2.1 cells (Fig. 1h) was repeated in more than 2000 independent transfections.

The membrane trafficking of JEDI-1P in dissociated neurons (Fig. 2k) was repeated in three independent batches of neurons with independent transfections.

The micrographs of JEDI-1P and mCherry expression, JEDI-1P and tdTomato expression, and JEDI-1P only expression were repeated in 4, 3, and 2 mice, respectively.

## Reporting summary
Further information on research design is available in the Nature Portfolio Reporting Summary linked to this article.

## Data availability
All source data are provided with this paper in the Source Data file and the Github repository. All raw data generated in this study are available upon request. The sequence of JEDI-1P is available from GenBank (accession number OP342601 [https://ncbi.nlm.nih.gov/nuccore/2476883238]). Plasmids used for in vitro characterization and packaging AAV for in vivo voltage imaging are available from addgene (accession numbers are pc3-puro-CAG-JEDI-1P-P2A-mCherry-CAAX, 202606; pc3-puro-CAG-JEDI-1P, 202607; pAAV-EF1a-DIO-JEDI-1P-Kv, 202608; pAAV-EF1a-DIO-mCherry, 202620; pAAV-EF1a-DIO-tdTomato, 202610). Data used to generate all figures are provided in the Source Data file. Data used to generate in vivo imaging related figures is also available at https://github.com/JaegerLab/JEDI-1P_Widefiled_Figures_data[91]. Source data are provided with this paper.

## Code availability
The code for in vivo imaging pre-processing pipeline is available at https://github.com/JaegerLab/JEDI-1P-Kv_widefield_imaging_preprocessing_pipeline[71]. The code for generating in vivo imaging related figures is available at https://github.com/JaegerLab/JEDI-1P_Widefiled_Figures_data[91].

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

## Acknowledgements

The authors acknowledge Dr. Benjamin Arenkiel, Joshua Ortiz-Guzman, and Zihong Chen at the Intellectual and Developmental Disabilities Research Center Neuroconnectivity Core for AAV packaging; this Core is supported by NIH grant P50HD103555 and the Charif Souki Fund. We thank Jason Kirk, Hannah Johnson and the Optical Imaging & Vital Microscopy (OiVM) Core at Baylor College of Medicine (BCM) for assistance with the confocal microscopy and the plate reader. We thank Shuyuan Yang (St-Pierre lab) for helping image neurons by confocal microscopy. We thank Francisco Alejandro Blanco, Dr. Christopher Asberry Cronkite, and Dr. Joseph Gerald Duman from Dr. Kimberley Renee Fuchs Tolias lab (BCM) for preparing neurons for in vitro characterization. We thank Dr. Junzhan Jing and Dr. Xiaolong Jiang at BCM for sharing voltage waveforms recorded from pyramidal neurons and Dr. Indira Monica Raman at Northwestern University for sharing voltage waveforms recorded from Purkinje neurons. We acknowledge Dr. Shella Keilholz from Emory University and Dr. Arthur Morrissette for helping with the in vivo imaging analyses. We thank Madison Cohen, Ellie Jiayi He, and Brune Le Chatelier at Emory University for helping with animal handling. The project was supported by the Klingenstein-Simons Fellowship Award in Neuroscience (F.S.-P.); the McNair Medical Foundation (F.S.-P.); Welch Foundation grants Q-2016-20190330 and Q-2016-20220331 (F.S.-P.); a John S. Dunn Collaborative Research Award (F.S.-P.); NIH grants R01NS111470 (D.J.), S10OD016244 (D.J.), R01EB027145 (F.S.-P.), U01NS113294 (F.S.-P.), U01NS118288 (F.S.-P.), R01EB032854 (F.S.P), and RF1NS128901 (F.S.-P.); NSF grants 1707359 and 1935265 (F.S.-P.) and Udall grant P50NS123103 (D.J.).

## Author contributions

D.J. and F.S.P. conceived and oversaw the project. X.L., Y.W., Z.L., D.J., and F.S.P. prepared figures and wrote the manuscript. GEVI screening and in vitro characterization: Z.L. developed the screening platform hardware; X.L. and Z.L. developed the screening platform software and data analysis code; X.L. and Y.G. screened GEVIs; X.L. characterized GEVIs in vitro and cloned viral vectors. In vivo imaging: Y.W. conducted all in vivo imaging experiments and developed code for pre-processing pipeline and related analyses.

## Competing interests

The authors declare the following competing interest: FSP is an inventor on a US patent (#US9606100 B2) that encompasses the design and specific uses of voltage indicators that share the same architecture as the JEDI-1P indicator discussed in this article. Leland Stanford Junior University is the patent applicant and current assignee. Co-inventor MZ Lin is not an author on this paper. There are no other competing interests.
