## [Peer Review File · Nature Communications]

REVIEWER COMMENTS

Reviewer #1 (Remarks to the Author):

The paper by Lu et al describes the development of a new member of the ASAP GEVI family. The GEVI has higher voltage sensitivity, faster kinetics, and better photostability compared with older versions. The authors demonstrated its usage for widefield voltage imaging which allowed the detection of high-frequency spontaneous and evoked population events across the mouse cortex.

The protein engineering part of the paper is impressive, and the new indicator seems a valuable addition to the GEVI arsenal. While the widefield application is useful, similar performance could have been likely achieved with older GEVIs which were already demonstrated to generate similar data using fiber photometry (e.g. Marshall et al, Cell, 2016 and Kannan et al, Nat Methods, 2018). Thus, I'm not sure that the authors fully demonstrated the potential of this indicator. I overall support this paper, below I have a few suggestions that I believe could strengthen the paper.

1) As said, the new indicator seems very good and I believe that it could be useful for more demanding applications such as voltage imaging at cellular resolution. The only cellular resolution data provided in this paper is from engineered HEK cells triggered by AP-like waveforms. It would be thus very informative if the authors would provide data from real neurons. In vivo recordings (e.g. from cortical L1) would be ideal, but in vitro data from either cultured neurons or brain slices would be also fine. Showing that this indicator has superior performance in neurons compared with ASAP3 in parameters such as (1) spike sensitivity and SNR, (2) ability to detect high-frequency firing, and (3) photostability under the light intensities needed for cellular resolution voltage imaging would be very helpful.

2) To be compatible with complex behavioral tasks, imaging should be ideally continuous. The authors decided to break their imaging sessions into trials, likely due to photostability issues, but it is not explained well. Please describe in the main text the exact imaging protocol, i.e. for how long the brain was imaged continuously and how many trials were recorded per day. Adding a time bar to figure 3 indicating the on-off timing could also help the reader understand the protocol. Furthermore, please provide information about the maximum possible duration of continuous recording. These data would help readers to assess the applicability of this technology to their experiments.

3) Related to the previous comment, in figure 2 the authors provide photostability data for only 2 minutes, could you please provide stability information for longer periods (at least 10 minutes)?

Minor comment:

The JEDI-1P point is missing from figure 1d-e (it does appear on panel 'f').

Reviewer #2 (Remarks to the Author):

Nat Comm ref report

Lu et al. develop a new genetically-encoded voltage sensor and apply it to in vivo imaging of mouse neocortex demonstrating apparently useful signals. The new sensor JEDI-1P appears brighter and faster than ASAP2 and ASAP3.

Major comment:

For the in vivo mouse imaging, the authors should show substantially more data to convince the reader of the utility of the methodology. The authors should show image time series to indicate the spatiotemporal structure of fluorescence signals for both spontaneous and evoked activity. Previous wide-field imaging using organic dyes revealed propagating waves of activity and sensory-evoked activity spreading from obvious focal points in sensory cortex (e.g. Ferezou et al., 2007). Does the current methodology offer similar dynamic views of the neocortex. This is not obvious from the current presentation of data. I suggest the authors show extensive examples of raw image series, and how the processing pipeline changes these signals, again as image time series.

Minor comments:

1. Fig 1d,e,f - Why is JEDI-1P part of panel f in Figure 1, and not d and e?
2. Fig 2d - The command voltage for the 'action potential' seems rather long. It might be interesting to compare responses to different AP waveforms, including fast ones (mimicking PV neurons) and slower ones (mimicking excitatory neurons).
3. Fig. 2f - What is the 'width' for the actual voltage command waveform used?
4. Fig. 3a - The ICV transfection method might create bias in which populations of neurons are more strongly labelled. Later-developing excitatory neurons and various types of GABAergic neurons might be differentially infected, as well as other AAV tropisms.

5. Fig. 3 and in vivo work - I do not understand why the authors chose to use two AAV vectors to deliver JEDI-1P and mCherry/tdTomato separately? Why not use the P2A single vector, on which the biophysical tests were made, if I understand correctly. The two vector strategy could lead to differential infection across different neurons and brain areas.

6. Fig. 3 and in vivo work - Different AAV vector combinations were used. AAV9 vs pHp capsids. What worked better?

7. Fig. 3/4 - using RFP to remove artefacts from the JEDI signal might be good, but relies on the two having common bleaching rates, and similar modulation to artefacts like breathing, heart beat, light scattering changes, etc... This is not really demonstrated in the current study, and indeed it is likely that this methodology can only be partially successful. Red vs Green fluorescence will be differentially absorbed, scattered and affected by the various artefacts. The authors might comment on the rationale and limits of their methodology.

8. Fig. 4 - It appears to be important to remove various artefacts from the fluorescence signals, and the approach of selectively removing RFP fluctuations in various frequency bands appears promising. I think it would be important to show image sequences in this figure, so that the reader can understand how the pipeline affects time series. How do the blood vessels appear?

9. Fig. 4d - I am surprised to see the large slow hemodynamic component apparently dominating fluorescence changes in the awake condition. Is that typical? More details, examples and image time-series would be valuable to show. Is there a danger that some of these slow signals reflect real electrical responses in the neuronal tissue, and that important aspects of cortical function are being discarded in the processing pipeline? Whole-cell recordings reveal substantial slow Vm dynamics, even in awake mice.

10. Figure 5 - How was the LFP filtered? Is it a DC signal? I don't think isoflurane is ideal for these experiments, because it generates the very sharp brief transients also shown in Fig. 5d. Perhaps the authors could carry out some awake experiments, or use a different anaesthetic, like ketamine/xylazine or urethane, which tend to generate slower cortical dynamics?

11. Figure 5 - It would seem obvious to try to correlate the fluorescence signals with membrane potential measurements in vivo. Although not essential for this publication, the authors could carry out whole-cell recordings to correlate single-cell membrane potential with network activity reported

through fluorescence, as originally carried out by Ferezou, Grinvald, Petersen and Sakmann in the investigation of organic voltage-sensitive dyes in awake head-restrained mice.

12. Figure 6 - The reader needs to see raw df signals represented as image series over time. I think it would be great if the authors could do a well-controlled single whisker stimulus, studying the initiation of the signal and its spread over time across the cortex.

13. Table 1 - Tau (slow) repolarisation - no number given for JEDI-1P. Fraction fast - no number given for JEDI-1P. Why?

14. Why were awake mice water restricted?

-

Yours sincerely,

Carl Petersen.

19th September 2022, EPFL.

Reviewer #3 (Remarks to the Author):

Dear editor,

I have read the recent submission by Lu et al describing a new ASAP GEVI named JEDI-1P. On a positive note, this new GEVI yields a fast optical response with a larger $\Delta F/F$ than ASAP 3. However, there are several issues with this submission that prevents me from recommending publication at this time.

Major issue.

There is no 2-Photon data. Normally, I would not ask for more experiments, but given that the authors have just published a JEDI-2P probe that differs by only two amino acids from this GEVI, I must insist on a 2P comparison.

I must also insist on a comparison in this paper of JEDI-1P with JEDI-2P with 1P microscopy. The author's state that they did not do this since JEDI-2P behaves like ASAP3 which is not satisfactory. Indeed, I believe that there is a difference in the hyperpolarizing signal as well as in the so-called photon budget. In the supplemental data to the JEDI-2p paper (see of Liu et al., 2022 Sustained deep-tissue voltage recording using a fast indicator evolved for two-photon microscopy). JEDI-2P behaves better (has a faster off than ASAP 3).

ASAP1 and ASAP2s exhibited an activity dependent rundown (see Chamberland et al., 2017 Fast two-photon imaging of subcellular voltage dynamics in neuronal tissue with genetically encoded indicators Figure 3). The authors claim that the decay in ASAP1 and ASP2s mimic the circuit, however the optical activity of ArcLight and GCaMP6f suggested otherwise. The JEDI-2P also shows persistent activity (see figure 4 of Liu et al., 2022 Sustained deep-tissue voltage recording using a fast indicator evolved for two-photon microscopy). The authors should show how JEDI-1P behaves in a similar fashion to see if JEDI-1P has an activity induced rundown as well.

Photon budgets. The authors used a different light intensity than previously reported I guess because of figure 2i. Please report signal to noise comparisons. Figure 2i suggests to me that the authors should have used even more intense light. The fact that they did not suggests that more intense light did not help. This raises another concern about bleaching. Figure 2g shows relative bleaching rates for illumination with 2.6 mW/mm² 470 nm light. However, the recordings were done with 34 mW/mm². That is very misleading. Please report the bleaching using the stronger light intensity.

Other issues

Line 79 claiming that circular permuted FP have the potential to produce larger signals I believe is incorrect. The signals are different but the size may be larger for FRET probes versus cpFPs or vice versa. Please provide a theoretical explanation for that claim.

Line 99. Am I to understand that multiparametric means the authors tried long voltage pulses in addition to short voltage pulses? It seems to me the authors are claiming that long pulses were as informative as short pulses. So why do short pulses? You can still see the speed of the response regardless of length of the pulse.

Line 107. As expected, the GEVI/mCherry ratio showed less variation from well to well compared with GEVI (green) intensity values alone.

This is very problematic. Are the authors suggesting that co-expression with mCherry improved GEVI expression (reduced variability)? How is that possible? If I divide a number by an integer larger than 1 of course the result will be a reduction but that does not mean that the variability of the GEVI expression has changed.

Line 130 the authors claim that JEDI-2P gave response 174% and 71% larger than ASAP2 and ASAP 3 respectively. That is an extremely misleading claim. The authors should simply state the dynamic range of each construct without this dramatic attempt to inflate the response.

For the cortical expression, the authors added a targeting motif from Kv2.1. They should show the voltage-dependent optical properties of that construct as well to ensure that the probe is behaving in a similar fashion as the original JEDI-1P.

Line 202. It was unclear to me what was being imaged. Is it spontaneous activity? I also do not understand the claimed correlation with 60 Hz activity. What is causing the 60 Hz activity in the LFP? Where is the optical activity of JEDI-1P that correlates at 60 Hz. I read the claim but do not see the evidence.

Figure 1j. The authors should comment on the extremely broad excitation spectrum of JEDI-1P. Can the authors detect a voltage-dependent signal with excitation at 400 nm? Is that protonation of the chromophore? The authors should test the effect of pH given the large shoulder at 390 nm.

Figure 2d. Why is the baseline shifted after the GEVIs respond to the command voltage replicates several action potentials? It is not bleaching as the baseline is stable afterwards.

Figure 6d. The high frequency response is not impressive. Nor are the low frequency responses for that matter as no baseline is presented for comparison. Where is the signal?

Figure 6i and j. What are the dashed lines? I thought they indicated air puffs but then why does the last air puff not elicit a response in both 6i and 6j?

Line 273. I do not think 'exquisitely' is appropriate as ASAP3 looks to be better attuned to hyperpolarization as well as 20 mV depolarization (figure 2a).

Line 276. I have not seen any evidence of JEDI-1p reporting 60 Hz oscillations in the cortex.

Reviewers' comments are in black. We have addressed reviewers's comment in blue and indicated substantive changes with blue highlighting in the manuscript

REVIEWER #1

The paper by Lu et al describes the development of a new member of the ASAP GEVI family. The GEVI has higher voltage sensitivity, faster kinetics, and better photostability compared with older versions. The authors demonstrated its usage for widefield voltage imaging which allowed the detection of high-frequency spontaneous and evoked population events across the mouse cortex.

The protein engineering part of the paper is impressive, and the new indicator seems a valuable addition to the GEVI arsenal.

Thank you!

While the widefield application is useful, similar performance could have been likely achieved with older GEVIs which were already demonstrated to generate similar data using fiber photometry (e.g. Marshall et al, Cell, 2016 and Kannan et al, Nat Methods, 2018). Thus, I'm not sure that the authors fully demonstrated the potential of this indicator. I overall support this paper, below I have a few suggestions that I believe could strengthen the paper.

Thank you for supporting the paper and for your suggestions.

1) As said, the new indicator seems very good and I believe that it could be useful for more demanding applications such as voltage imaging at cellular resolution. The only cellular resolution data provided in this paper is from engineered HEK cells triggered by AP-like waveforms. It would be thus very informative if the authors would provide data from real neurons. *In vivo* recordings (e.g. from cortical L1) would be ideal, but *in vitro* data from either cultured neurons or brain slices would be also fine. Showing that this indicator has superior performance in neurons compared with ASAP3 in parameters such as (1) spike sensitivity and SNR, (2) ability to detect high-frequency firing, and (3) photostability under the light intensities needed for cellular resolution voltage imaging would be very helpful.

We thank the Reviewer for the kind words and the suggestion. We agree that additional comparisons with ASAP3 would be valuable to the community. From the choices suggested by the Reviewer, we selected to do experiments in dissociated neurons since these are standard in the field. We also expanded our characterization *in vivo* using the paradigm described in our paper, i.e., widefield cortical imaging. Given the large number of experiments the three reviewers suggested, we could not provide data with concurrent patch-clamping and voltage imaging *in vivo* within a reasonable timespan. Here is a summary of the results:

Request #1: Spike sensitivity and SNR. JEDI-1P generated significantly larger responses and SNR to single APs in dissociated neurons than and ASAP3 (Fig. 2m-o, Fig. S5b).

Request #2: Evaluate the ability of JEDI-1P to detect high-frequency firing. JEDI-1P can track high-frequency action potentials recorded from Purkinje cells in a single trial. JEDI-1P also displays larger responses and higher spike-detection accuracy than ASAP2s and ASAP3 (Fig. S3c-e)

Request #3: Photostability under the light intensities needed for cellular resolution voltage imaging. We cannot address this query because *in vivo* 1P imaging of ASAP/JEDIs in mice with cellular resolution has not yet been reported. Such an application is challenging due to background fluorescence from neurons above and below the plane of focus. Demonstrating 1P cellular voltage imaging in mice is beyond the scope of this publication. Moreover, the mapping between *in vitro* and *in vivo* irradiances is complicated because light scattering and absorption by the brain will drastically reduce the irradiance at the plane of the target neuron. The irradiance at the focal plane thus depends on imaging depth.

Despite these concerns, we acknowledge the desire of the Reviewer for more data on 1P photostability. We provide the following new data.

(#3.1) Since our paper is focused on widefield cortical imaging, we provided an expanded characterization *in vivo* under this imaging modality. We included new data showing that JEDI-1P has outstanding photostability during 1

hour of continuous illumination (**Fig. 3i**). We also used schematics to clarify the description of our *in vivo* photostability characterization experiments (**Fig. 3f-i**).

(#3.2) We show photobleaching traces (as done in **Fig. 2g**) for the full range of irradiances described in **Fig. 2i (Fig S4)**. These experiments were done in HEK293-Kir2.1, our cell line with neuronal-like polarized resting membrane potential. These cells are efficiently transfected, enabling more rapid data collection. We reiterate that the photobleaching rates *in vivo* using the same power are expected to be drastically reduced due to light absorption and scattering.

2) To be compatible with complex behavioral tasks, imaging should be ideally continuous. The authors decided to break their imaging sessions into trials, likely due to photostability issues, but it is not explained well. Please describe in the main text the exact imaging protocol, i.e. for how long the brain was imaged continuously and how many trials were recorded per day. Adding a time bar to figure 3 indicating the on-off timing could also help the reader understand the protocol. Furthermore, please provide information about the maximum possible duration of continuous recording. These data would help readers to assess the applicability of this technology to their experiments.

We thank the Reviewer for allowing us to clarify this point. Imaging sessions were broken into trials because the Scimedia Ultima software that was supplied with our setup by the vendor cannot record for longer than 200 s and takes a long time to save the large files between trials. We added this sentence to the legend of Fig.3: "Data was recorded in separate trials due to software limitations with continuous data acquisition."

To address the Reviewer's concern, we performed continuous illumination *in vivo* for 1 hour. We only captured images at 0, 30, and 60 min to avoid the software's image acquisition limitations. We found that JEDI-1 conserves $95.8 \pm 4.2\%$ over this duration (**Fig. 3i**). These results suggest that photostability is sufficient for multi-hour experiments. As requested, we have clarified the imaging protocol in the main text and included time bars in **Fig. 3f-i**.

3) Related to the previous comment, in figure 2 the authors provide photostability data for only 2 minutes, could you please provide stability information for longer periods (at least 10 minutes)?

Fig. 2g now reports photostability over 10 minutes.

Minor comment:

The JEDI-1P point is missing from figure 1d-e (it does appear on panel 'f').

Thank you for highlighting this discrepancy. Panels d-f should show the performance of the platform before we started screening. We mistakenly included JEDI-1P in Fig. 1f. We removed this data point in the revised manuscript. In case reviewers would like to see the correlations, including JEDI-1P, we have included the same graphs with JEDI-1P in **Fig. S1 e-g**.

REVIEWER #2

Lu et al. develop a new genetically-encoded voltage sensor and apply it to in vivo imaging of mouse neocortex demonstrating apparently useful signals. The new sensor JEDI-1P appears brighter and faster than ASAP2 and ASAP3.

Thank you for the positive comments.

Major comment:

For the in vivo mouse imaging, the authors should show substantially more data to convince the reader of the utility of the methodology. The authors should show image time series to indicate the spatiotemporal structure of fluorescence signals for both spontaneous and evoked activity. Previous widefield imaging using organic dyes revealed propagating waves of activity and sensory-evoked activity spreading from obvious focal points in sensory cortex (e.g. Ferezou et al., 2007). Does the current methodology offer similar dynamic views of the neocortex. This is not obvious from the current presentation of data. I suggest the authors show extensive examples of raw image series, and how the processing pipeline changes these signals, again as image time series.

We thank the Reviewer for this valuable suggestion. A new panel (**Fig. 6b**) now shows the requested image time series during a single whisker deflection, where we see activity spreading from contralateral barrel cortex following the stimulus and quickly diminishing after 35 ms. Interestingly, we see clear changes within 10 ms imaging windows, supporting the value of fast (200 Hz) imaging.

As requested, we provided raw imaging time series and showed how the processing pipeline changes signals changes these signals (**Fig. S9**).

Minor comments:

1. Fig 1d,e,f - Why is JEDI-1P part of panel f in Figure 1, and not d and e?

See the response to reviewer #1, minor comment.

2. Fig 2d - The command voltage for the 'action potential' seems rather long. It might be interesting to compare responses to different AP waveforms, including fast ones (mimicking PV neurons) and slower ones (mimicking excitatory neurons).

The command voltage waveform we chose for the action potential was recorded from a representative hippocampal neuron and resampled to 2-ms FWHM to mimic the shape of a layer 2/3 cortical neurons at room temperature (Hedrick and Waters 2012), since *in vitro* voltage-clamp experiments in **Fig. 2a-f** were done at room temperature.

As suggested, we performed a voltage-clamp experiments with simulated action potential waveforms from narrower (0.25-ms FWHM) to wider (4 ms). These experiments were conducted at 33C, a more physiological temperature (seals are difficult to maintain at 37C). JEDI-1P produced larger responses to spikes of all widths than ASAP2s and ASAP3 (**Fig. S3a,b**). We also characterized GEVIs' responses to voltage waveforms recorded from Purkinje neurons (**Fig. S3, c-e**). We showed that JEDI-1P detects high-frequency firing of narrow action potentials with higher precision and enables single-trial detection of simulated APs up to 300Hz.

3. Fig. 2f - What is the 'width' for the actual voltage command waveform used?

2-ms, as indicated in the legend to **Fig. 2d**, where these waveforms are first introduced.

4. Fig. 3a - The ICV transfection method might create bias in which populations of neurons are more strongly labelled. Later-developing excitatory neurons and various types of GABAergic neurons might be differentially infected, as well as other AAV tropisms.

Indeed, the neonatal ICV injection in EMX1-Cre transgenic line limits the expression of JEDI-1P-Kv to only excitatory neurons present at the time of the injection. We have also injected JEDI-1P-Kv in a VGAT-Cre line to express the sensor in interneurons. While there was successful expression in inhibitory neurons, the expression was mainly in deeper layers. Our imaging window preparation where the skull remains intact might not be best suited for imaging deeper layers. Future development of JEDI-1P transgenic mouse lines will enable the

expression of JEDI-1P in cell types not present at birth. Retro-orbital injection as an AAV delivery method could work in adult mice. However, in our experience, the expression of fluorescence using retro-orbital injection was more sparse and less uniform and is ultimately cost-prohibitive.

To address the Reviewer's comment, we added the following sentence in the discussion:

“Because neonatal ICV injection does not label neurons that develop at later stages, the cell population visualized differs from preparations where vectors are administered to adult mice.”

5. Fig. 3 and in vivo work - I do not understand why the authors chose to use two AAV vectors to deliver JEDI-1P and mCherry/tdTomato separately? Why not use the P2A single vector, on which the biophysical tests were made, if I understand correctly. The two vector strategy could lead to differential infection across different neurons and brain areas.

We could not use this strategy due to AAV packaging limit constraints (we are at ~4.5 kb and the AAV packaging limit is ~4.7 kb). GEVI-P2A-RFP was previously tested via lipofection of plasmids in dissociated neurons (*unpublished experiments*). However, the resulting transfected neurons showed poor localization of GEVIs at the membrane and many fluorescent puncta. The precise origin of this phenotype is not entirely clear but may be due to significant readthrough (rather than ribosome skipping) at the 2A peptide.

Expressing a reference fluorescence was done to capture non-voltage signals such as hemodynamics and motion artifacts. While the two vectors had a differential infection pattern, the preprocessing pipeline minimized the impact of the differences in expression level at different cell types. First, filtering reference signals at different frequencies allowed flexible separation of artifacts. Second, the preprocessing pipeline uses Ordinary Least-Squares Regression to scale the reference signal to the same level as the JEDI-1P signal at a given frequency range before regression. The regression was also performed locally at binned pixels, which was less impacted by differential expressions at different brain regions.

6. Fig. 3 and in vivo work - Different AAV vector combinations were used. AAV9 vs pHp capsids. What worked better?

Both combinations achieved similarly robust expression of JEDI-1P and the reference fluorescence and enabled imaging of neural signals at the cortical level. We have added the previous sentence to the methods.

7. Fig. 3/4 - using RFP to remove artefacts from the JEDI signal might be good, but relies on the two having common bleaching rates, and similar modulation to artefacts like breathing, heart beat, light scattering changes, etc... This is not really demonstrated in the current study, and indeed it is likely that this methodology can only be partially successful. Red vs Green fluorescence will be differentially absorbed, scattered and affected by the various artefacts. The authors might comment on the rationale and limits of their methodology.

Thank you for raising these concerns. The preprocessing pipeline corrects the JEDI-1P signal for photobleaching prior to regression (**Fig. 4c**). mCherry RFP was excited with the same light as JEDI-1P (466/40). We did not detect photobleaching of mCherry (**Fig. 3i**) and therefore did not need to correct its signal.

We agree that green and red light will be differentially scattered and absorbed, and on average, red light will travel slightly longer before being scattered. These differences will lead to red and green light being modulated slightly differently by breathing and hemodynamic artifacts. We have added this sentence to the discussion:

“Because green and red light are differentially scattered and absorbed in tissue, breathing and hemodynamic artifacts may modulate these wavelengths slightly differently, making it challenging to eliminate these artifacts entirely.”

8. Fig. 4 - It appears to be important to remove various artefacts from the fluorescence signals, and the approach of selectively removing RFP fluctuations in various frequency bands appears promising. I think it would be important to show image sequences in this figure, so that the reader can understand how the pipeline affects time series.

Thanks for the suggestion. We have added image sequences in **Fig. 6b** and **S9**.

How do the blood vessels appear?

Blood vessels obscure fluorescence and neural activity. Not surprisingly, they produce a lower correlation with LFP recordings (e.g., Fig. S10c, f).

9. Fig. 4d - I am surprised to see the large slow hemodynamic component apparently dominating fluorescence changes in the awake condition. Is that typical?

Slow hemodynamic signals are quite variable from trial to trial, and they can result in positive or negative signal corrections. It is typical that they can be quite large. These observations are consistent with previous observations of slow changes in arteriole diameter, as reviewed by Kleinfeld and colleagues (doi: [10.1016/j.neuron.2020.07.020](https://doi.org/10.1016/j.neuron.2020.07.020))

More details, examples and image time-series would be valuable to show.

We now show both raw and processed image sequences (Fig. S9).

Is there a danger that some of these slow signals reflect real electrical responses in the neuronal tissue, and that important aspects of cortical function are being discarded in the processing pipeline? Whole-cell recordings reveal substantial slow Vm dynamics, even in awake mice.

The reference channel is fluorescence from a non-voltage-sensitive RFP. The slow hemodynamic captured by the reference fluorescence thus do not include voltage signals. Since regression is performed using a scaled signal from the reference, our pipeline is not expected to remove voltage signals. Further, hemodynamic signals linked to membrane depolarization typically have a hemodynamic delay of at least 1s. The delay makes it unlikely that a waveform present without any delay in both the reference and JEDI-1P channels is related to voltage.

A larger dataset (not part of this paper) shows slow voltage signal dynamics in voltage after preprocessing (below), demonstrating that these components are not systematically removed. The figure shows a global ramping activity before the delivery of water reward in 6 task-performing mice. We ran a temporal-independent component analysis on 1 s averaged activity from the exposed cortex for each animal and extracted six independent components. Subpanel (a) shows consistent ramping activity in all mice (only one IC is shown per mouse), and the spatial maps show the weights of corresponding IC. Subpanel (b) shows the original signal from different ROIs and reconstructed signals using 6 ICs, both showing the global ramping activity.

As mentioned above, our new Fig. S9, which shows raw image sequences and processed image sequences, can also help to clarify our process.

10. Figure 5 - How was the LFP filtered? Is it a DC signal?

LFP signal went through a notch filter at 60 Hz. During analysis, LFP signal was also filtered using a lowpass filter at 70 Hz to match the filter applied to the optical signals. While this was previously mentioned in the captions, we added this information to our methods

I don't think isoflurane is ideal for these experiments, because it generates the very sharp brief transients also shown in Fig. 5d. Perhaps the authors could carry out some awake experiments, or use a different anaesthetic, like ketamine/xylazine or urethane, which tend to generate slower cortical dynamics?

As the Reviewer suggested, the wide-spread large oscillations observed in anesthetized experiments were induced by isoflurane. Irrespective of the source of the large oscillations, the anesthetized experiment in Figure 5b showed correlation and minimal latency between optical signal and electrophysiological signals. We performed simultaneous imaging and LFP recording in awake mice, which showed local correlation coefficient between LFP recording and optical recording (**Fig. S10**). Trials with air-puff stimulation also showed coherence up to 60 Hz and its spatial distribution.

11. Figure 5 - It would seem obvious to try to correlate the fluorescence signals with membrane potential measurements *in vivo*. Although not essential for this publication, the authors could carry out whole-cell recordings to correlate single-cell membrane potential with network activity reported through fluorescence, as originally carried out by Ferezou, Grinvald, Petersen and Sakmann in the investigation of organic voltage-sensitive dyes in awake head-restrained mice.

We agree that whole-cell recording *in vivo* simultaneous with voltage imaging could provide additional information, but it is beyond the scope of this study. We appreciate that the reviewer did not consider this experiment to be essential for this publication.

12. Figure 6 - The reader needs to see raw df signals represented as image series over time. I think it would be great if the authors could do a well-controlled single whisker stimulus, studying the initiation of the signal and its spread over time across the cortex.

We thank the Reviewer for the helpful suggestion. **Figure 6** now includes imaging data during single whisker stimulation, which showed fast response at contralateral barrel cortex within 5 ms (**Fig. 6b-c**) as well as the spread of the signals across the cortex (**Fig. 6b**).

13. Table 1 - Tau (slow) repolarisation - no number given for JEDI-1P. Fraction fast - no number given for JEDI-1P. Why?

We tried fitting the off kinetics of JEDI-1P with both mono-exponential and double-exponential function. With double exponential fitting, the fast component and the slow component were very close, and the percent of fast component was close to 50%, suggesting the off-kinetics of JEDI-1P may be better fitted with a mono exponential than a double exponential.

We added a note to Table 1 to clarify this point.

14. Why were awake mice water restricted?

Being head-fixed for up to an hour can be stressful for the animals. Water reward during the long-period recording minimized motion artifacts caused by anxiety. To maximize the data collected, we also incorporated behavioral tasks into the imaging when we tested photostability of the sensor instead of recording spontaneous activity for hundreds of hours. These behavioral data are not part of this study.

-

Yours sincerely,

Carl Petersen.

19th September 2022, EPFL.

REVIEWER #3

Dear editor,

I have read the recent submission by Lu et al describing a new ASAP GEVI named JEDI-1P. On a positive note, this new GEVI yields a fast optical response with a larger $\Delta F/F$ than ASAP 3.

Thank you

However, there are several issues with this submission that prevents me from recommending publication at this time.

Major issue.

There is no 2-Photon data. Normally, I would not ask for more experiments, but given that the authors have just published a JEDI-2P probe that differs by only two amino acids from this GEVI, I must insist on a 2P comparison.

We conducted the experiment suggested by the Reviewer and tested JEDI-1P under two-photon microscopy. While JEDI-1P and JEDI-2P have similar response amplitude and brightness under 2P excitation, JEDI-1P is less photostable than JEDI-2P (**Fig. S6**). These results validate our approach to develop indicators optimized for specific imaging modalities, and we recommend using JEDI-1P for 1P and JEDI-2P for 2P.

We thank the Reviewer for not requesting 2P *in vivo* experiments, since these would be outside the scope of our manuscript on 1-photon widefield cortical imaging.

I must also insist on a comparison in this paper of JEDI-1P with JEDI-2P with 1P microscopy. The author's state that they did not do this since JEDI-2P behaves like ASAP3 which is not satisfactory. Indeed, I believe that there is a difference in the hyperpolarizing signal as well as in the so-called photon budget. In the supplemental data to the JEDI-2p paper (see of Liu et al., 2022 Sustained deep-tissue voltage recording using a fast indicator evolved for two-photon microscopy). JEDI-2P behaves better (has a faster off than ASAP 3).

We thank the Reviewer for this valuable suggestion. The Reviewer's comment prompted us to do a more systematic characterization of JEDI-2P under 1P. Our experiments revealed a decrease in JEDI-2P's responses to voltage changes over time under widefield 1P illumination (**Fig. S5**). These results suggest that continuous 1P illumination produces fluorescent but non-voltage-responsive indicator molecules. We speculate that JEDI-2P's T207H mutation, also present in photoactivatable GFP (Patterson and Lippincott-Schwartz 2002), may be responsible for this phototransformation. As a result, while a full exploration of this effect is beyond the scope of the current study, we recommend using JEDI-1P for voltage imaging under 1P.

The above considerations are presented in the Discussion. Note that the above behavior is specific to 1P illumination, as JEDI-2P produces consistent responses under 2P over prolonged (>30 min) excitation (see our 2022 Cell paper).

ASAP1 and ASAP2s exhibited an activity dependent rundown (see Chamberland et al., 2017 Fast two-photon imaging of subcellular voltage dynamics in neuronal tissue with genetically encoded indicators Figure 3). The authors claim that the decay in ASAP1 and ASP2s mimic the circuit, however the optical activity of ArLight and GCaMP6f suggested otherwise. The JEDI-2P also shows persistent activity (see figure 4 of Liu et al., 2022 Sustained deep-tissue voltage recording using a fast indicator evolved for two-photon microscopy). The authors should show how JEDI-1P behaves in a similar fashion to see if JEDI-1P has an activity induced rundown as well.

We thank the reviewer for giving us the opportunity to clarify data in previous papers. In Chamberland et al., 2017, we show that ASAP1 and ASAP2s —when expressed in L2 cells of the fly visual circuit and monitored using 2P microscopy— produce transient responses upon changes in contrast.

In other words, the responses of ASAP1 and ASAP2s have a significant τ_{decay} (see graph below).

As stated in the Chamberland et al., these findings are “consistent with electrophysiological recordings in lamina monopolar cells (Zettler and Järvilehto, 1971, <https://doi.org/10.1007/BF00630560>) and our prior GEVI-imaging experiments (Yang et al., 2016, <https://doi.org/10.1016/j.cell.2016.05.031>)”. The Reviewer correctly points out that these transient responses are not seen in ArcLight and GCaMP6f. In the same paper, we hypothesize this is due to the much slower kinetics of ArcLight and the calcium indicator GCaMP6f. Responses to step voltages in these indicators show no desensitization (Fig. 1D of Chamberland et al.), suggesting that the transient responses mimic the underlying voltage. However, since we do not have simultaneous electrophysiological recordings in axon terminals (which is not achievable), we cannot directly confirm our hypotheses.

In our recent JEDI-2P paper, changes in the visual stimulus (e.g., its coverage in the visual field) resulted in a change in the response phenotype (see below). This change was observed with previous variants such as ASAP2f and ASAP2s. It is thus not a property of the indicator, consistent with responses to step depolarizations in vitro.

Taken together, we do not believe that there is a concern with the 2P fly data shown in previous papers. Repeating these experiments in flies with JEDI-1P under two-photon microscopy is outside the scope of this manuscript.

Photon budgets. The authors used a different light intensity than previously reported I guess because of figure 2i. Please report signal to noise comparisons. Figure 2i suggests to me that the authors should have used even more intense light. The fact that they did not suggests that more intense light did not help. This raises another concern about bleaching. Figure 2g shows relative bleaching rates for illumination with 2.6 mW/mm² 470 nm light. However, the recordings were done with 34 mW/mm². That is very misleading. Please report the bleaching using the stronger light intensity.

Thanks for raising the concern for photostability under different illumination power and we appreciate you giving us a chance to explain the rationale for these technical details.

Indeed, we performed photostability characterization under different irradiances. We chose to present the photobleaching trace under 2.6 mW/mm² illumination because, within the range tested, it is the closest to that

used in vivo experiments (0.05 mW/mm²). Therefore, we consider it more informative for readers interested in widefield imaging in mice, the focus of the current paper. (We did not evaluate photostability under lower irradiances given that JEDI-1P already shows excellent photostability at this irradiance level). We added a sentence to the main text to help clarify our choice of irradiance level.

We agree that other users may be interested in different (higher) irradiances. Data at different irradiances was already presented (**Fig. 2i**). To address the Reviewer's concern, we included photobleaching traces under higher irradiances (**Fig. S4**).

Other issues

1. Line 79 claiming that circular permuted FP have the potential to produce larger signals I believe is incorrect. The signals are different but the size may be larger for FRET probes versus cpFPs or vice versa. Please provide a theoretical explanation for that claim.

Since a comparison of the theoretical advantages of FRET vs cpFP-based indicators is not important for our manuscript, we removed this controversial sentence. (NB: we like FRET-based indicators too!)

2. Line 99. Am I to understand that multiparametric means the authors tried long voltage pulses in addition to short voltage pulses? It seems to me the authors are claiming that long pulses were as informative as short pulses. So why do short pulses? You can still see the speed of the response regardless of length of the pulse.

We thank the Reviewer for the comment. First, we would like to clarify that the multiparametric screening refers to measuring multiple parameters (aka "performance metrics") in the same screen, including response amplitude (both to short and long pulses), brightness, and photostability. We clarified this in the final sentence of the 2nd paragraph of the result section.

We agree with the Reviewer that, in theory, GEVI response kinetics could be determined by fitting the slope of the responses at the onset (on-kinetics) and offset (off-kinetics) of the long voltage steps. However, such fits are extremely challenging and, in our hands, very noisy. In contrast, measuring the peak and width of the response to short voltage steps relatively simple.

3. Line 107. As expected, the GEVI/mCherry ratio showed less variation from well to well compared with GEVI (green) intensity values alone.

This is very problematic. Are the authors suggesting that co-expression with mCherry improved GEVI expression (reduced variability)? How is that possible? If I divide a number by an integer larger than 1 of course the result will be a reduction but that does not mean that the variability of the GEVI expression has changed.

We apologize for any confusion that our statement may have caused, and we appreciate the opportunity to clarify this point.

Even when the same construct is transfected in multiple wells, the observed GFP intensities can vary due to differences in the numbers of cells present in each well's field of view, in addition to pipetting errors affecting the amount of plasmid and transfection reagents used. These variations will also affect an RFP co-expressed with the GEVI through a P2A sequence. Normalizing the GFP signal with the RFP intensity will thus mitigate these variations. In **Fig. S1c**, we describe variability using the Coefficient of Variation (CoV), which is calculated by dividing the standard deviation by the mean. This approach avoids producing "mathematical artifacts," as discussed by the Reviewer, since dividing the GFP values by the same number will reduce both the standard deviation and mean by the same factor, leading to the same CoV.

4. Line 130 the authors claim that JEDI-2P gave response 174% and 71% larger than ASAP2 and ASAP 3 respectively. That is an extremely misleading claim. The authors should simply state the dynamic range of each construct without this dramatic attempt to inflate the response.

We thank the Reviewer for pointing that our phrasing could lead to confusion. We took the Reviewer's suggestion to state the dynamic range of each construct. We kept the relative changes, which we believe are helpful and no longer misleading in the revised context. The revised phrasing is now:

“JEDI-1P exhibited a response of $-31.2 \pm 2.0\%$ (mean \pm 95% CI here and henceforth) to spike waveforms, equivalent to a 174% increase over the response of ASAP2s ($-11.4 \pm 0.8\%$) and a 71% increase over the response of ASAP3 ($-18.3 \pm 2.0\%$) (Fig. 2d-e).”

5. For the cortical expression, the authors added a targeting motif from Kv2.1. They should show the voltage-dependent optical properties of that construct as well to ensure that the probe is behaving in a similar fashion as the original JEDI-1P.

We have previously shown that the Kv2.1 tag had no or minimal impact on the voltage sensitivity of ASAP3, a GEVI closely related to JEDI-1P (Villette et al., Cell, 2019). We show this figure (Fig. S6 from our 2019 paper) below. This figure shows that ASAP3 and ASAP3-Kv have response vs. voltage curves that are nearly identical, with the possible exception of hyperpolarizations to -120 mV, which is not physiological for typical cortical neurons.

We realize that the Reviewer may be concerned that the improvements of JEDI-1P compared with ASAP3 may not extend to their variants with Kv2.1PRC tag. Our new experiments show that JEDI-1P-Kv produces larger and higher-SNR fluorescence transients than ASAP3-Kv in response to APs in cortical neurons (**Fig. 2m-o**).

6. Line 202. It was unclear to me what was being imaged. Is it spontaneous activity?

Yes, Figure 5 showed spontaneous activity under light anesthesia.

I also do not understand the claimed correlation with 60 Hz activity. What is causing the 60 Hz activity in the LFP?

This is spontaneous activity, for example due to oscillations and irregular firing of neurons.

Where is the optical activity of JEDI-1P that correlates at 60 Hz. I read the claim but do not see the evidence.

Figure 5 showed spontaneous activity under light anesthesia. The blue trace in **Fig 5e** indicated coherence to up to >50 Hz above shuffled control level between LFP and optical recordings at ROI 1i, which was adjacent to the recording site. The coherence was expected to be local rather than global. Since we processed LFP signals using a 60-Hz notch filter, coherence was not observed at exactly 60 Hz. We thus revised replaced “60 Hz” with “50 Hz” in the sentence:

“The closest site analyzed to the LFP recording showed coherence up to ~50 Hz (...)”

We now also added a new set of experiments with simultaneous awake optical-LFP recording with air-puff recording up to 60 Hz (**Fig. S10**). The coherence maps between awake LFP and optical recording showed coherence during each stimulation period up to 60 Hz.

It is worth noting that LFP recording and optical recording do not capture identical signals, which is why correlation or coherence close to 1 is not expected. LFP is mixed with local potentials and volume conductance from different sites, whereas our widefield voltage imaging using JEDI-1P-Kv mainly captures voltage changes from the soma and proximal dendrites.

7. Figure 1j [correction: 2j]. The authors should comment on the extremely broad excitation spectrum of JEDI-1P.

We appreciate the Reviewer for bringing up this astute observation. Because autofluorescence produces a substantial contribution in that region, results are acutely sensitive to the precise level of autofluorescence

subtraction. We refined our experimental procedure to more precisely subtract background by using an automated cell counter to precisely match cell numbers between JEDI-1P-expressing and control wells. We found that spectra with inaccurate autofluorescence correction could be identified as those with large deviations at wavelengths that should produce negligible fluorescence, based on the excitation spectrum of the (non-permuted) GFP of JEDI-1P. We therefore excluded spectra with excitation efficiency beyond $\pm 10\%$ excitation efficiency at 350 or 535 nm. The revised JEDI-1P excitation spectrum displays a reduced blue shoulder (**Fig. 2j**).

Can the authors detect a voltage-dependent signal with excitation at 400 nm?

We conducted new experiments to address the Reviewer's question. At ~ 400 nm, JEDI-1P displays a small bright-to-dim response as shown in the figure below (**Fig. S2f-h**). These results are consistent with a shift from deprotonated to protonated chromophores upon depolarization.

The authors should test the effect of pH given the large shoulder at 390 nm. Is that protonation of the chromophore?

A key mechanism of GFP-based indicators is modulation of the chromophore protonation. As a result, indicators are typically pH sensitive, e.g., (Helassa et al. 2016) (Wang et al. 2022) (Kang et al. 2019; Jin et al. 2012). As suggested by the Reviewer, we characterized the spectrum and the response amplitude of JEDI-1P under pH ranging from 6.6 to 8.0 (**Fig. S2a-e**). While JEDI-1P had a robust response at all pH, the amplitude of its peak and steady-state responses varied with pH, as expected. The height of the 'shoulder' near 400 nm increases at lower pH, suggesting that it is possibly the protonated form of the chromophore, as the Reviewer insightfully suggested.

We do not believe that sensitivity to pH is a large concern for most applications. For example, Theparambil and colleagues wrote that “the brain extracellular pH remains remarkably stable, not only withstanding variable levels in neuronal activity and metabolism, but also major (physiological or pathological) perturbations of systemic acid/base balance” (Nat. comm., 2021). Other papers support these conclusions e.g., a 5-s 60-Hz stimulation of the SN/VTA in rats produced changes in pH of 0.02-0.03 units (Venton et al., 2003; <https://doi.org/10.1046/j.1471-4159.2003.01527.x>).

8. Figure 2d. Why is the baseline shifted after the GEVIs respond to the command voltage replicates several action potentials? It is not bleaching as the baseline is stable afterwards.

We thank the reviewers for the keen observation and the opportunity to clarify. This is due to slow off-kinetics (aka kinetics to repolarization). ASAP2s and ASAP3 are slower than JEDI-1P, so their fluorescence did not recover to baseline by the time the next spike waveform was applied. This is discussed in the following sentence in the main text:

“Due to its faster kinetics, JEDI-1P fluorescence returned to baseline between individual spikes of a train of action potential waveforms (Fig. 2d) (...).

9. Figure 6d. The high frequency response is not impressive. Nor are the low frequency responses for that matter as no baseline is presented for comparison. Where is the signal?

Figure 6d was used to demonstrate that the neural response to high-frequency stimuli can be detected. The changes are relative to baseline intensity. The decreased amplitude of the oscillatory responses as stimulus frequency increased is not surprising because the reduced duration of each cycle is expected to reduce the amplitude of the resulting neural responses. The reduction in response amplitude at higher stimulus frequency is not due to an indicator limitation since JEDI-1P kinetics exceed the stimulus frequency.

The oscillatory responses following the large onset response (**Fig. 6l,m**) were also likely dominated by subthreshold activity rather than action potentials, which is another reason why high amplitude oscillations were not expected here. Previous work using LFP recording during oscillatory stimulation has demonstrated a similar response pattern in the barrel cortex (Bessaih et al. 2018).

10. Figure 6i and j. What are the dashed lines? I thought they indicated air puffs but then why does the last air puff not elicit a response in both 6i and 6j?

(Fig. 6i,j are now Fig. 6l,m in the revised manuscript)

We thank the Reviewer for pointing out the confusion with the last dashed line. The dashed lines are intended to show the digital trigger for opening of the valve and, by approximation, the onset of each air puff. The last dashed line was intended to show the end of the stimulation, which is now removed to avoid confusion. note that the latency between the dashed lines and the peaks of each oscillation cycle was mainly a result of the opening time of the valve (about 7 ms) but not the sensor. The revised manuscript includes the following sentence in the figure caption:

“Neural responses are delayed compared with the valve opening triggers (dashed lines) due to the ~7-ms valve opening time sensor.”

11. Line 273. I do not think ‘exquisitely’ is appropriate as ASAP3 looks to be better attuned to hyperpolarization as well as 20 mV depolarization (figure 2a).

We thank the reviewers for their careful reading of the manuscript. This paragraph was not on comparing JEDI-1P to ASAP3 but a discussion on the types of signals reported in our imaging paradigm. Nevertheless, we rephrased the beginning of this paragraph.

Of note, JEDI-1P and ASAP3 have the same response amplitude to 20-mV depolarizations (See Supplementary document “Statistical information”). Given that JEDI-1P has higher brightness, it will produce larger SNR to 20-mV depolarizations.

12. Line 276. I have not seen any evidence of JEDI-1p reporting 60 Hz oscillations in the cortex.

The revised manuscript includes a spatial map of power at 60 Hz in Figure 6e and 6i to illustrate that widefield voltage imaging captured 60 Hz activity. The updated subpanels show averaged data from 4-5 mice.

Fig. 6g and a new figure (**Fig. 6c**) highlight the power of responses up to 60 Hz in a single trial. Given the reduced response amplitudes to higher frequency stimuli, the power at 60 Hz was lower but visible in a single trial. To better illustrate that the 60 Hz response was present, the mean spectrograms of the responses to air-puff and flicker, respectively, are shown below and in **Fig. S12**. Power at 60 Hz can be identified in both spectrograms (Spectrogram for flicker: N = 40 trials, 4 mice, 10 trials/animal; spectrogram for air-puff response: N = 50 trials, 5 mice, 10 trials/animal).

REVIEWERS' COMMENTS

Reviewer #1 (Remarks to the Author):

The authors thoroughly addressed the reviewers' comments, I have no further requests.

Reviewer #2 (Remarks to the Author):

I think the authors have appropriately revised their manuscript.

-

Yours sincerely,

Carl Petersen

EPFL, 7th May 2023

Reviewer #3 (Remarks to the Author):

Dear editor,

I am not impressed with the advancements made in the JEDI-1P GEVI. I do not find the cortical imaging that informative, and I am certain other GEVIs would perform similarly. But I do not feel it is my role to determine what should be published in Nature Communications. My role is to critically assess the claims made. Whether this probe is actually useful is unclear, but does appear to be an improvement of ASAP3. I believe this is more in line with Scientific Reports, but if the editors believe otherwise I wouldn't reject it.

Below are my original comments (in black). The authors' responses are in blue, and my new comments are in red. I hope that both the editors and the authors' will read these comments because the GEVI field is full of misleading reports resulting in a great deal of wasted effort and resources.

I must also insist on a comparison in this paper of JEDI-1P with JEDI-2P with 1P microscopy. The author's state that they did not do this since JEDI-2P behaves like ASAP3 which is not satisfactory. Indeed, I believe that there is a difference in the hyperpolarizing signal as well as in the so-called photon budget. In the supplemental data to the JEDI-2p paper (see of Liu et al., 2022 Sustained deep-tissue voltage recording using a fast indicator evolved for two-photon microscopy). JEDI-2P behaves better (has a faster off than ASAP 3).

We thank the Reviewer for this valuable suggestion. The Reviewer's comment prompted us to do a more systematic characterization of JEDI-2P under 1P. Our experiments revealed a decrease in JEDI-2P's responses to voltage changes over time under widefield 1P illumination (Fig. S5). These results suggest that continuous 1P illumination produces fluorescent but non-voltage-responsive indicator molecules. We speculate that JEDI-2P's T207H mutation, also present in photoactivatable GFP (Patterson and Lippincott-Schwartz 2002), may be responsible for this phototransformation. As a result, while a full exploration of this effect is beyond the scope of the current study, we recommend using JEDI-1P for voltage imaging under 1P. The above considerations are presented in the Discussion. Note that the above behavior is specific to 1P illumination, as JEDI-2P produces consistent responses under 2P over prolonged (>30 min) excitation (see our 2022 Cell paper).

I could not access the supplementary data, but I believe the potential for rundown. Other ASAP probes have this so it is not surprising the JEDI probes also suffer from it.

ASAP1 and ASAP2s exhibited an activity dependent rundown (see Chamberland et al., 2017 Fast two-photon imaging of subcellular voltage dynamics in neuronal tissue with genetically encoded indicators Figure 3). The authors claim that the decay in ASAP1 and ASP2s mimic the circuit, however the optical activity of ArcLight and GCaMP6f suggested otherwise. The JEDI-2P also shows persistent activity (see figure 4 of Liu et al., 2022 Sustained deep-tissue voltage recording using a fast indicator evolved for two-photon microscopy). The authors should show how JEDI-1P behaves in a similar fashion to see if JEDI-1P has an activity induced rundown as well.

We thank the reviewer for giving us the opportunity to clarify data in previous papers. In Chamberland et al., 2017, we show that ASAP1 and ASAP2s —when expressed in L2 cells of the fly visual circuit and monitored using 2P microscopy— produce transient responses upon changes in contrast. In other words, the responses of ASAP1 and ASAP2s have a significant tdecay (see graph below).

I thank the authors for sharing their ASAP2s and JEDI-2P data (for some reason that report compared JEDI-2P to ASAP2f. I can see why they did not compare it to ArcLight.)

As stated in the Chamberland et al., these findings are “consistent with electrophysiological recordings in lamina monopolar cells (Zettler and Järvilehto, 1971, <https://doi.org/10.1007/BF00630560>) and our prior GEVI-imaging experiments (Yang et al., 2016, <https://doi.org/10.1016/j.cell.2016.05.031>)”.

I am aware of that statement. And after reviewing that data and the JEDI-2P data, it is clear that statement is wrong.

The Reviewer correctly points out that these transient responses are not seen in ArcLight and GCaMP6f.

It is also not seen in the JEDI-2P trace. I was under the impression that JEDI-2P was faster than ASAP2S (to quote the JEDI-2P paper, figure 2D-F ‘(D–F) JEDI-2P produces larger and faster responses to a spike waveform under 2PM than ASAP3 and ASAP2s.’ The JEDI-2P trace looks much more like the ArcLight trace than ASAP2s. Given that the authors have admitted that JEDI-1P runs down in 2P (and that JEDI-2P runs down under 1P), I am more likely to believe the ArcLight data and the JEDI-2P trace since they corroborate one another.

Also, I strongly encourage the authors to zero all of the optical traces so that it doesn’t appear like there is an attempt to make ASAP2s look better by starting ArcLight below the baseline. I am certain that was not the intention, but once it is published it is difficult to defend.

In the same paper, we hypothesize this is due to the much slower kinetics of ArcLight and the calcium indicator GCaMP6f. Responses to step voltages in these indicators show no desensitization (Fig. 1D of Chamberland et al.), suggesting that the transient responses mimic the underlying voltage. However, since we do not have simultaneous electrophysiological recordings in axon terminals (which is not achievable), we cannot directly confirm our hypotheses. In our recent JEDI-2P paper, changes in the visual stimulus (e.g., its coverage in the visual field) resulted in a change in the response phenotype (see below). This change was observed with previous variants such as ASAP2f and ASAP2s. It is thus not a property of the indicator,

Actually, it is. There is a rundown in ASAP1 and ASAP2s that is not seen in JEDI-2P or ArcLight.

consistent with responses to step depolarizations in vitro. Taken together, we do not believe that there is a concern with the 2P fly data shown in previous papers. Repeating these experiments in flies with JEDI-1P under two-photon microscopy is outside the scope of this manuscript.

A new GEVI is developed and an experiment done in their previous reports comparing ASAP1, ASAP2f, ASAP2s, MacQ-mcitrine, Ace2N-AA-mNeon, and JEDI-2P is not done for JEDI-1P?! JEDI-1P must perform extremely poorly under 2P conditions.

Other issues

1. Line 79 claiming that circular permuted FP have the potential to produce larger signals I believe is incorrect. The signals are different but the size may be larger for FRET probes versus cpFPs or vice versa. Please provide a theoretical explanation for that claim.

Since a comparison of the theoretical advantages of FRET vs cpFP-based indicators is not important for our manuscript, we removed this controversial sentence. (NB: we like FRET-based indicators too!)

2. Line 99. Am I to understand that multiparametric means the authors tried long voltage pulses in addition to short voltage pulses? It seems to me the authors are claiming that long pulses were as informative as short pulses. So why do short pulses? You can still see the speed of the response regardless of length of the pulse. We thank the Reviewer for the comment.

First, we would like to clarify that the multiparametric screening refers to measuring multiple parameters (aka “performance metrics”) in the same screen, including response amplitude (both to short and long pulses), brightness, and photostability. We clarified this in the final sentence of the 2nd paragraph of the result section. We agree with the Reviewer that, in theory, GEVI response kinetics could be determined by fitting the slope of the responses at the onset (on-kinetics) and offset (off-kinetics) of the long voltage steps. However, such fits are extremely challenging and, in our hands, very noisy. In contrast, measuring the peak and width of the response to short voltage steps relatively simple.

This is a rather concerning response. If the recording is too noisy to fit to a simple exponential decay during an in vitro voltage clamp experiment, why would should the scientific community invest the time and effort to measure its activity in vivo?

4. Line 130 the authors claim that JEDI-2P gave response 174% and 71% larger than ASAP2 and ASAP 3 respectively. That is an extremely misleading claim. The authors should simply state the dynamic range of each construct without this dramatic attempt to inflate the response.

We thank the Reviewer for pointing that our phrasing could lead to confusion. We took the Reviewer’s suggestion to state the dynamic range of each construct. We kept the relative changes, which we believe are helpful and no longer misleading in the revised context. The revised phrasing is now: “ JEDI-1P exhibited a response of $-31.2 \pm 2.0\%$ (mean \pm 95% CI here and henceforth) to spike waveforms,

equivalent to a 174% increase over the response of ASAP2s ($-11.4 \pm 0.8\%$) and a 71% increase over the response of ASAP3 ($-18.3 \pm 2.0\%$) (Fig. 2d-e).”

Technically, this is only true if the resting light levels are the same for all three probes. However, I think this is a much more honest presentation. I thank the authors for this change.

7. Figure 1j [correction: 2j]. The authors should comment on the extremely broad excitation spectrum of JEDI-1P.

We appreciate the Reviewer for bringing up this astute observation. Because autofluorescence produces a substantial contribution in that region, results are acutely sensitive to the precise level of autofluorescence subtraction. We refined our experimental procedure to more precisely subtract background by using an automated cell counter to precisely match cell numbers between JEDI-1P-expressing and control wells. We found that spectra with inaccurate autofluorescence correction could be identified as those with large deviations at wavelengths that should produce negligible fluorescence, based on the excitation spectrum of the (non-permuted) GFP of JEDI-1P. We therefore excluded spectra with excitation efficiency beyond $\pm 10\%$ excitation efficiency at 350 or 535 nm. The revised JEDI-1P excitation spectrum displays a reduced blue shoulder (Fig. 2j).

Can the authors detect a voltage-dependent signal with excitation at 400 nm?

We conducted new experiments to address the Reviewer’s question. At ~ 400 nm, JEDI-1P displays a small bright-to-dim response as shown in the figure below (Fig. S2f-h). These results are consistent with a shift from deprotonated to protonated chromophores upon depolarization.

I thank the authors for doing that experiment, however, their conclusion is incorrect. That result is not consistent with a shift from protonation to deprotonation. Again, I apologize for not being able to see the supplemental data, but I believe the authors did see a voltage-dependent decrease for the 400 nm excitation experiment.

It is a reasonable hypothesis that the voltage-dependent signal involves the protonation state of the chromophore. Indeed, that was the reason for suggesting the experiment. However, if both the 470 nm excitation and 400 nm excitation both get dimmer during depolarization of the plasma membrane, then it definitely does not involve a change in the protonation state. For example, if the voltage transient during the 470nm illumination experiment which excites the deprotonated form of the chromophore caused the chromophore to become protonated then there would be a decrease in the fluorescence emitted, but there should also be a corresponding increase in the 400 nm excitation since the population of the protonated chromophore increased. The authors report that the probe dims during 470 and 400 nm excitation effectively ruling out that hypothesis.

8. Figure 2d. Why is the baseline shifted after the GEVIs respond to the command voltage replicates several action potentials? It is not bleaching as the baseline is stable afterwards.

We thank the reviewers for the keen observation and the opportunity to clarify. This is due to slow off-kinetics (aka kinetics to repolarization). ASAP2s and ASAP3 are slower than JEDI-1P, so their fluorescence did not recover to baseline by the time the next spike waveform was applied. This is discussed in the

following sentence in the main text: “Due to its faster kinetics, JEDI-1P fluorescence returned to baseline between individual spikes of a train of action potential waveforms (Fig. 2d) (...).

I would ask that the author’s remove that sentence. Even JEDI-1P has a change in the baseline (though smaller than the other two). As it does not really affect this submission, I don’t think it needs to be addressed further. I was simply curious as to why.

Response to final review

Color code:

Black bolded = Reviewer's original comment

Blue = our original response to the comments

Red = the Reviewer's response to our response in blue

Green = our response to the Reviewer's comments in red. We indented these comments to help distinguish them

Dear editor,

I am not impressed with the advancements made in the JEDI-1P GEVI. I do not find the cortical imaging that informative, and I am certain other GEVIs would perform similarly. But I do not feel it is my role to determine what should be published in Nature Communications. My role is to critically assess the claims made. Whether this probe is actually useful is unclear, but does appear to be an improvement of ASAP3. I believe this is more in line with Scientific Reports, but if the editors believe otherwise I wouldn't reject it.

Below are my original comments (in black). The authors' responses are in blue, and my new comments are in red. I hope that both the editors and the authors' will read these comments because the GEVI field is full of misleading reports resulting in a great deal of wasted effort and resources.

I must also insist on a comparison in this paper of JEDI-1P with JEDI-2P with 1P microscopy. The authors state that they did not do this since JEDI-2P behaves like ASAP3 which is not satisfactory. Indeed, I believe that there is a difference in the hyperpolarizing signal as well as in the so-called photon budget. In the supplemental data to the JEDI-2p paper (see of Liu et al., 2022 Sustained deep- tissue voltage recording using a fast indicator evolved for two-photon microscopy). JEDI-2P behaves better (has a faster off than ASAP 3).

We thank the Reviewer for this valuable suggestion. The Reviewer's comment prompted us to do a more systematic characterization of JEDI-2P under 1P. Our experiments revealed a decrease in JEDI-2P's responses to voltage changes over time under widefield 1P illumination (Fig. S5). These results suggest that continuous 1P illumination produces fluorescent but non-voltage-responsive indicator molecules. We speculate that JEDI-2P's T207H mutation, also present in photoactivatable GFP (Patterson and Lippincott-Schwartz 2002), may be responsible for this phototransformation. As a result, while a full exploration of this effect is beyond the scope of the current study, we recommend using JEDI-1P for voltage imaging under 1P. The above considerations are presented in the Discussion. Note that the above behavior is specific to 1P illumination, as JEDI-2P produces consistent responses under 2P over prolonged (>30 min) excitation (see our 2022 Cell paper).

I could not access the supplementary data, but I believe the potential for rundown. Other ASAP probes have this so it is not surprising the JEDI probes also suffer from it.

[We do not provide a response here as there is no question or comment for us to reply

ASAP1 and ASAP2s exhibited an activity dependent rundown (see Chamberland et al., 2017 Fast two- photon imaging of subcellular voltage dynamics in neuronal tissue with genetically encoded indicators Figure 3). The authors claim that the decay in ASAP1 and ASP2s mimic the circuit, however the optical activity of ArcLight and GCaMP6f suggested otherwise. The JEDI-2P also shows persistent activity (see figure 4 of Liu et al., 2022 Sustained deep-tissue voltage recording using a fast indicator evolved for two-photon microscopy). The authors should show how JEDI-1P behaves in a similar fashion to see if JEDI-1P has an activity induced rundown as well.

We thank the Reviewer for giving us the opportunity to clarify data in previous papers. In Chamberland et al., 2017, we show that ASAP1 and ASAP2s —when expressed in L2 cells of the fly visual circuit and monitored using 2P

microscopy— produce transient responses upon changes in contrast. In other words, the responses of ASAP1 and ASAP2s have a significant τ decay (see graph below).

I thank the authors for sharing their ASAP2s and JEDI-2P data (for some reason that report compared JEDI-2P to ASAP-2f. I can see why they did not compare it to ArcLight.)

This manuscript focuses on 1P widefield cortical voltage imaging in mice. Since we do not report 2P voltage imaging in flies, a discussion of this concern is outside the scope of the current paper. For the benefit of this Reviewer, we can clarify that ASAP2f and ASAP2s produce comparable responses to alternating dark/light flashes in *Drosophila* L2 neurons. However, ASAP2f is faster than ASAP2s, as shown in their respective publications. Before the development of JEDI-2P, ASAP2f was thus the preferred GEVI for use in flies by the Clandinin lab. We thus chose ASAP2f as reference GEVI when benchmarking JEDI-2P in Liu et.al, 2022 (<https://doi.org/10.1016/j.cell.2022.07.013>).

As stated in the Chamberland et al., these findings are "consistent with electrophysiological recordings in lamina monopolar cells (Zettler and Järvillehto, 1971, <https://doi.org/10.1007/BF00630560>) and our prior GEVI-imaging experiments (Yang et al., 2016, <https://doi.org/10.1016/j.cell.2016.05.031>)".

I am aware of that statement. And after reviewing that data and the JEDI-2P data, it is clear that statement is wrong.

We respect (but disagree with) the Reviewer's position. Given that the comment refers to a statement from a 2017 paper, we do not believe this manuscript provides the right vehicle for a scholarly discussion of this prior publication. We encourage the Reviewer to explore their claim that other sensors that may provide higher-fidelity voltage readouts and produce a manuscript to share their findings with the community.

The Reviewer correctly points out that these transient responses are not seen in ArcLight and GCaMP6f.

It is also not seen in the JEDI-2P trace. I was under the impression that JEDI-2P was faster than ASAP2S (to quote the JEDI-2P paper, figure 2D-F '(D–F) JEDI-2P produces larger and faster responses to a spike waveform under 2PM than ASAP3 and ASAP2s.' The JEDI-2P trace looks much more like the ArcLight trace than ASAP2s. Given that the authors have admitted that JEDI-1P runs down in 2P (and that JEDI-2P runs down under 1P), I am more likely to believe the ArcLight data and the JEDI-2P trace since they corroborate one another

As previously stated, statements on other indicators in flies are outside the scope of this manuscript. We clarify what we did not state nor present evidence that JEDI-1P "runs down" in 2P, assuming we correctly interpret that the Reviewer defines "run down" as a change in the fractional response amplitude over time. What we reported is that JEDI-1P has faster photobleaching under 2P than JEDI-2P.

Also, I strongly encourage the authors to zero all of the optical traces so that it doesn't appear like there is an attempt to make ASAP2s look better by starting ArcLight below the baseline. I am certain that was not the intention, but once it is published it is difficult to defend.

Because this comment refers to a previous publication and is unrelated to the current manuscript, we will limit our discussion of this topic. We will only clarify that the horizontal black line in Figure 3 of Chamberland et al. (2017) is the mean fluorescence per trial, not the zero line. The response amplitude of each indicator is reflected via the vertical scale bars. The above is stated in the figure caption.

In the same paper, we hypothesize this is due to the much slower kinetics of ArcLight and the calcium indicator GCaMP6f. Responses to step voltages in these indicators show no desensitization (Fig. 1D of Chamberland et al.), suggesting that the transient responses mimic the underlying voltage. However, since we do not have simultaneous electrophysiological recordings in axon terminals (which is not achievable), we cannot directly confirm our hypotheses. In our recent JEDI-2P paper, changes in the visual stimulus (e.g., its coverage in the visual field) resulted

in a change in the response phenotype (see below). This change was observed with previous variants such as ASAP2f and ASAP2s. It is thus not a property of the indicator,

Actually, it is. There is a rundown in ASAP1 and ASAP2s that is not seen in JEDI-2P or ArCLight.

This comment has been addressed by our comments above, and it is outside the scope of the current manuscript on JEDI-1P-based widefield voltage imaging in mice.

consistent with responses to step depolarizations in vitro. Taken together, we do not believe that there is a concern with the 2P fly data shown in previous papers. Repeating these experiments in flies with JEDI-1P under two-photon microscopy is outside the scope of this manuscript

A new GEVI is developed and an experiment done in their previous reports comparing ASAP1, ASAP2f, ASAP2s, MacQ-mcitrine, Ace2N-AA-mNeon, and JEDI-2P is not done for JEDI-1P?! JEDI-1P must perform extremely poorly under 2P conditions.

The in vitro performance of JEDI-1P under 2P illumination is shown in Fig. S6. In this figure, we showed that under 2P illumination, JEDI-1P and JEDI-2P displayed similar response amplitude, brightness and kinetics compared to each other, while JEDI-2P is more photostable than JEDI-1P. This result was also clearly stated in the text (line 150-151: 'Under laser-scanning 2P, JEDI-1P displayed similar properties to JEDI-2P except for a lower photostability'; line 281-282: 'However, JEDI-1P was more photolabile than JEDI-2P under laser-scanning 2P microscopy').

We agree that the visual circuits processing in flies is an interesting topic and could be advanced by two-photon voltage imaging. However, since this manuscript focuses on reporting a 1P-optimized GEVI and its application in mice under 1P widefield imaging, we found repeating a 2P experiment in flies outside of the scope of this manuscript.

Other issues

1. Line 79 claiming that circular permuted FP have the potential to produce larger signals I believe is incorrect. The signals are different but the size may be larger for FRET probes versus cpFPs or vice versa. Please provide a theoretical explanation for that claim.

Since a comparison of the theoretical advantages of FRET vs cpFP-based indicators is not important for our manuscript, we removed this controversial sentence. (NB: we like FRET-based indicators too!)

2. Line 99. Am I to understand that multiparametric means the authors tried long voltage pulses in addition to short voltage pulses? It seems to me the authors are claiming that long pulses were as informative as short pulses. So why do short pulses? You can still see the speed of the response regardless of length of the pulse. We thank the Reviewer for the comment.

First, we would like to clarify that the multiparametric screening refers to measuring multiple parameters (aka "performance metrics") in the same screen, including response amplitude (both to short and long pulses), brightness, and photostability. We clarified this in the final sentence of the 2nd paragraph of the result section. We agree with the Reviewer that, in theory, GEVI response kinetics could be determined by fitting the slope of the responses at the onset (on-kinetics) and offset (off-kinetics) of the long voltage steps. However, such fits are extremely challenging and, in our hands, very noisy. In contrast, measuring the peak and width of the response to short voltage steps relatively simple.

This is a rather concerning response. If the recording is too noisy to fit to a simple exponential decay during an in vitro voltage clamp experiment, why would should the scientific community invest the time and effort to measure its activity in vivo?

Our previous statement refers to challenges with quantitatively evaluating the on-response kinetics of GEVIs (not limited to JEDI-1P) from field stimulation data during high-throughput screening. Our statement is unrelated to applying JEDI-1P *in vivo* for reporting voltage dynamics as changes in fluorescence. Our manuscript thoroughly benchmarks JEDI-1P for 1P widefield applications *in vivo*.

4. Line 130 the authors claim that JEDI-2P gave response 174% and 71% larger than ASAP2 and ASAP 3 respectively. That is an extremely misleading claim. The authors should simply state the dynamic range of each construct without this dramatic attempt to inflate the response.

We thank the Reviewer for pointing that our phrasing could lead to confusion. We took the Reviewer's suggestion to state the dynamic range of each construct. We kept the relative changes, which we believe are helpful and no longer misleading in the revised context. The revised phrasing is now: "JEDI- 1P exhibited a response of $-31.2 \pm 2.0\%$ (mean \pm 95% CI here and henceforth) to spike waveforms, equivalent to a 174% increase over the response of ASAP2s ($-11.4 \pm 0.8\%$) and a 71% increase over the response of ASAP3 ($-18.3 \pm 2.0\%$) (Fig. 2d-e)."

Technically, this is only true if the resting light levels are the same for all three probes. However, I think this is a much more honest presentation. I thank the authors for this change.

Thank you! Of note, under shot-noise-limited conditions, fractional fluorescence changes to a given voltage change should be independent of variations in the baseline fluorescence.

7. Figure 1j [correction: 2j]. The authors should comment on the extremely broad excitation spectrum of JEDI-1P.

We appreciate the Reviewer for bringing up this astute observation. Because autofluorescence produces a substantial contribution in that region, results are acutely sensitive to the precise level of autofluorescence subtraction. We refined our experimental procedure to more precisely subtract background by using an automated cell counter to precisely match cell numbers between JEDI-1P-expressing and control wells. We found that spectra with inaccurate autofluorescence correction could be identified as those with large deviations at wavelengths that should produce negligible fluorescence, based on the excitation spectrum of the (non-permuted) GFP of JEDI-1P. We therefore excluded spectra with excitation efficiency beyond $\pm 10\%$ excitation efficiency at 350 or 535 nm. The revised JEDI-1P excitation spectrum displays a reduced blue shoulder (Fig. 2j).

Can the authors detect a voltage-dependent signal with excitation at 400 nm?

We conducted new experiments to address the Reviewer's question. At ~ 400 nm, JEDI-1P displays a small bright-to-dim response as shown in the figure below (Fig. S2f-h). These results are consistent with a shift from deprotonated to protonated chromophores upon depolarization.

I thank the authors for doing that experiment, however, their conclusion is incorrect. That result is not consistent with a shift from protonation to deprotonation. Again, I apologize for not being able to see the supplemental data, but I believe the authors did see a voltage-dependent decrease for the 400 nm excitation experiment.

It is a reasonable hypothesis that the voltage-dependent signal involves the protonation state of the chromophore. Indeed, that was the reason for suggesting the experiment. However, if both the 470 nm excitation and 400 nm excitation both get dimmer during depolarization of the plasma membrane, then it definitely does not involve a change in the protonation state. For example, if the voltage transient during the 470nm illumination experiment which excites the deprotonated form of the chromophore caused the chromophore to become protonated then there would be a decrease in the fluorescence emitted, but there should also be a corresponding increase in the 400 nm excitation since the population of the protonated chromophore increased. The authors report that the probe dims during 470 and 400 nm excitation effectively ruling out that hypothesis.

We agree with the reviewers we did not accurately phrase the conclusion of these experiments because we missed the y-axis in SFig. 2g used negative values ($-\% \Delta F/F$). SFig. 2g shows that the fluorescence at ~400 nm dims down with depolarizations: the polarity of the response is thus the same when using either 400- or 470-nm light. We cannot rule out that protonation change is occurring as it could be obscured by other mechanisms that lower fluorescence at 400 nm upon depolarization, such as a decrease in the quantum yield. However, we agree with the Reviewer that these results are inconsistent with the hypothesis that protonation changes is the only mechanism at play.

8. Figure 2d. Why is the baseline shifted after the GEVIs respond to the command voltage replicates several action potentials? It is not bleaching as the baseline is stable afterwards.

We thank the reviewers for the keen observation and the opportunity to clarify. This is due to slow off- kinetics (aka kinetics to repolarization). ASAP2s and ASAP3 are slower than JEDI-1P, so their fluorescence did not recover to baseline by the time the next spike waveform was applied. This is discussed in the following sentence in the main text: "Due to its faster kinetics, JEDI-1P fluorescence returned to baseline between individual spikes of a train of action potential waveforms (Fig. 2d) (...).

I would ask that the author's remove that sentence. Even JEDI-1P has a change in the baseline (though smaller than the other two). As it does not really affect this submission, I don't think it needs to be addressed further. I was simply curious as to why.

In the main text, we rephrased the sentence as '...fluorescence returned closer to baseline between individual spikes of a train of action potential waveforms (Fig. 2d)'.